# Fibrous network nature of plant cell walls enables tunable mechanics for development

Si Chen [1,2,3], Isabella Burda[1,3,4,5], Purvil Jani [6], Bex Pendrak[1,2], Meredith N. Silberstein [1,2] ✉ & Adrienne H. K. Roeder [1,3,4] ✉

During plant development, the mechanical properties of the cell walls must be tuned to regulate the growth of the cells. Cell growth involves significant stretching of the cell walls, yet little is known about the mechanical properties of cell walls under such substantial deformation, or how these mechanical properties change to regulate development. Here, we investigated the mechanical behavior of the Arabidopsis leaf epidermal cells being stretched. We found that the mechanical properties arise from the cell wall, which behaves as a fibrous network material. The epidermis exhibited a non-linear stiffening behavior that fell into three regimes. Each regime corresponded to distinct nonlinear behaviors in terms of transverse deformation (i.e., Poisson effect) and unrecoverable deformation (i.e., plasticity). Using a model, we demonstrated that the transition from reorientation and bending-dominated to stretch-dominated deformation modes of cellulose microfibrils cause these nonlinear behaviors. We found the stiffening behavior is more pronounced at later developmental stages. Finally, we show the *spiral2-2* mutant has anisotropic mechanical properties, likely contributing to the spiraling of leaves. Our findings reveal the fibrous network nature of cell walls gives a high degree of tunability in mechanical properties, which allows cells to adjust these properties to support proper development.

Recent studies have highlighted the critical roles that mechanics play in plant growth and morphogenesis[1–7]. The primary plant cell wall (referred to as "cell wall" hereafter) acts as a mechanically robust shell enclosing cells[8]. It physically constrains cell growth[9–11], while also influencing force patterns within tissues[6], which are increasingly recognized as instructive signals regulating biological activities[12–14]. In the epidermis (the outermost cell layer), cell walls are typically thicker and stiffer, serving as a mechanical growth constraint against inner cells and playing a crucial role in plant morphogenesis[15,16]. During development, cells adjust growth rates and directions to form proper organ shapes[17]. This process requires adapting cell wall mechanics to developmental needs. Thus, understanding the

mechanics of epidermal cell walls, especially in the context of development, is essential.

The cell wall consists of cellulose microfibrils embedded in a hydrated matrix of pectin and hemicellulose[8]. Cellulose microfibrils, synthesized by cellulose synthase complexes guided by cortical microtubules, serve as the primary load-bearing components due to their high stiffness[10]. The detailed architecture of cell walls is complex and significant progress has been made in recent years to understand it[18]. Studies using atomic force microscopy and electron tomography on onion epidermal cell walls suggest a cross-lamellate structure, where cellulose microfibrils are similarly oriented within each lamella but shift stepwise between adjacent lamellae[19,20]. However, whether

[1]Engineered Living Materials Institute, Cornell University, Ithaca, NY, USA. [2]Sibley School of Mechanical and Aerospace Engineering, Cornell University, Ithaca, NY, USA. [3]Weill Institute for Cell and Molecular Biology, Cornell University, Ithaca, NY, USA. [4]Section of Plant Biology, School of Integrative Plant Science, Cornell University, Ithaca, NY, USA. [5]Genetics, Genomics, and Development Graduate Program, Cornell University, Ithaca, NY, USA. [6]School of Chemical and Biomolecular Engineering, Cornell University, Ithaca, NY, USA. ✉e-mail: meredith.silberstein@cornell.edu; ahr75@cornell.edu

cell walls in other species have the same structure remains unclear. For example, studies on Arabidopsis roots imply a smoother, continuous rotation of cellulose microfibrils[21]. In cell wall architecture, an important concept is the "biomechanical hotspot," which serves as a load-bearing junction between cellulose microfibrils, though more characterization on this hypothetical junction is needed[18,22].

Plant cell growth occurs through the physical expansion of the cell wall driven by turgor pressure[9,23]. While depositing new wall material prevents wall thinning, it typically does not directly drive growth[24]. Wall extensibility (the ability of the cell wall to irreversibly expand) and anisotropy (mechanical properties varying with direction) influence growth rate and direction, respectively[10]. While anisotropy is associated with the orientation distribution of cellulose microfibrils within the cell wall, it remains unclear which aspects of cell wall architecture control extensibility[24]. In addition, the cell wall can go through tremendous stretching during growth. Therefore, it is crucial to investigate how cell walls are being stretched largely under in-plane forces, both reversibly and irreversibly, and understand the underlying deformation mechanism, possibly giving insights into cell wall architecture.

Utilizing approaches in the engineering discipline to study cell wall mechanics can improve our understanding of the relationship between cell wall architecture and mechanical behavior. Firstly, the mechanical properties of cell walls must be measured through carefully designed tests to accurately quantify applied forces and deformations[25]. One fundamental property in engineering studies is Poisson's ratio, which describes lateral deformation during stretching[26]. This property is often overlooked in cell wall studies, despite its importance in understanding material nanostructure and deformation mechanisms. Secondly, to connect nanostructure with mechanical behavior, it is important to focus on features that significantly impact macroscopic behavior. In plant cell walls, cellulose microfibrils are recognized as the primary load-bearing elements, leading to their historically being modeled as fiber-reinforced materials (the soft matrix is reinforced with stiff fibers)[27–31]. However, recent studies emphasize the importance of connections between cellulose microfibrils (e.g., biomechanical hotspots, cellulose-cellulose contacts), which form a connected fiber network[32]. This perspective aligns with the emerging concept of fibrous network materials (e.g., mammalian skin) in engineering and physics[33–35]. Such materials show intriguing mechanical behavior, such as strain-stiffening (stiffness increase as it deforms) and stiffening tunability (strain-stiffening response can be tuned), which arises from the collective deformation of fibers, influenced by the network structure and fiber-fiber interactions[36,37]. This mechanism should be considered when interpreting the mechanical behavior of cell walls.

*Arabidopsis thaliana*, a key model organism in plant development, has been extensively used to understand plant development from genetic and biochemical perspectives[38]. However, understanding of Arabidopsis cell wall mechanics under stretching remains limited. Two types of mechanical tests are commonly used to measure the mechanical properties of plants[39,40]: (1) indentation-based tests, where an indenter is used to poke the sample while measuring the force and displacement[41]; (2) tensile tests, which involve stretching the sample and monitoring force and deformation changes[42–44]. Most mechanical tests done on Arabidopsis are indentation tests due to their feasibility on small-size samples[45,46]. However, there are concerns about how accurately these tests relate to in-plane mechanical properties, which are particularly relevant for growth[10,40,47]. In addition, indentation tests often require assumptions, such as the mechanical behavior of the cell wall or the cell geometry, to interpret the data, which may influence the reliability of the results[48,49]. In contrast, tensile tests can directly determine mechanical properties such as stiffness, Poisson's ratio, anisotropy, and plasticity without complex interpretive processes. However, few tensile tests have been done on Arabidopsis plants[50–53],

and these tests have several limitations: (1) They are typically performed at the organ scale, like hypocotyl or stem, making it difficult to relate the results to cell wall properties due to the complex geometry of cell patterning. (2) They usually focus on small deformation regions. However, the importance of understanding nonlinear mechanical behavior across a large deformation regime has recently been recognized[47,54]. (3) Existing studies typically focus on the relationship between deformation in the loading direction and force, neglecting Poisson's ratio. (4) These tensile properties are not studied within the context of plant development. Onion epidermal cell walls are commonly used to study tensile mechanical properties because the specimen preparation procedure is well established[42–44,47,54]. However, it is unclear whether these walls are still actively growing, and they may not fully represent typical growing cell walls. Therefore, it might be difficult to relate their mechanical properties directly to growth.

In this study, we stretched the Arabidopsis leaf epidermis at different developmental stages on a micromechanical tensile testing stage coupled with confocal microscopy. The Arabidopsis leaf epidermis exhibited a nonlinear mechanical behavior that naturally fell into three regimes. By adapting a five-beam model, we found that these nonlinear behaviors result from the transition of cellulose microfibrils from reorientation and bending-dominated to stretch-dominated deformation modes. We observed that stiffening behavior is enhanced during development as growth slows, likely due to changes in the connections between cellulose microfibrils. This highlights that the cell wall behaves like a fibrous network material. Our work offers a more complete picture of how cell walls deform to resist turgor pressure and how mechanical properties are tuned for plant development.

## Results

### Nonlinear mechanical behavior of leaf epidermal cell layers

During growth, the cell wall undergoes tremendous deformation. To explore the mechanical properties associated with such a large deformation range, we stretched the epidermal peels of first pair of true leaves from Arabidopsis (Fig. 1a) using a tensile stage (Fig. 1b; Supplementary Methods 1.2). Each peel has one layer of abaxial (bottom side of the leaf) intact epidermal cells, which comprises both the upper and lower periclinal walls as well as anticlinal walls (Supplementary Fig. 1a–b; Supplementary Methods 1.5 and 2). We cut a thin rectangular strip for testing, with the long side parallel to the midrib and positioned approximately midway between the midrib and the leaf margin (Fig. 1A). To accurately quantify the force and deformation of the epidermal peel test samples, we conducted the tensile tests on a confocal microscope to optically measure deformation by tracking fluorescent beads[55] (Supplementary Methods 1.4) coating the outside of the tissue (Fig. 1b–c; Supplementary Movie 1).

To understand how the epidermal tissue resists deformation under stretching, we performed a monotonic tensile test, in which the sample was stretched at a constant rate until it failed (Supplementary Movie 1). The force per width (force divided by the width of the sample, also known as membrane force in mechanics; Supplementary Methods 1.6) versus axial stretch (deformed length of the sample divided by the initial length of the sample, a convention commonly used in large-deformation mechanics) curve reveals three regimes (Fig. 1d). To determine the stiffness in these regimes, we examined the tangent stiffness (slope of the force per width versus axial stretch) (Fig. 1e). In regime I, the sample is relatively soft, with a nearly constant low level of stiffness (~ 12 N/m, which is equivalent to 12 MPa if an effective wall thickness of 1 micron is assumed, based on the outer periclinal wall being about 1 micron thick[56] and the inner periclinal wall being too thin to be considered[57]). Then in regime II, the stiffness gradually increases as the stretch increases, exhibiting a nonlinear strain-stiffening behavior. This mechanical behavior suggests that more "elements" within the sample are becoming active in resistance

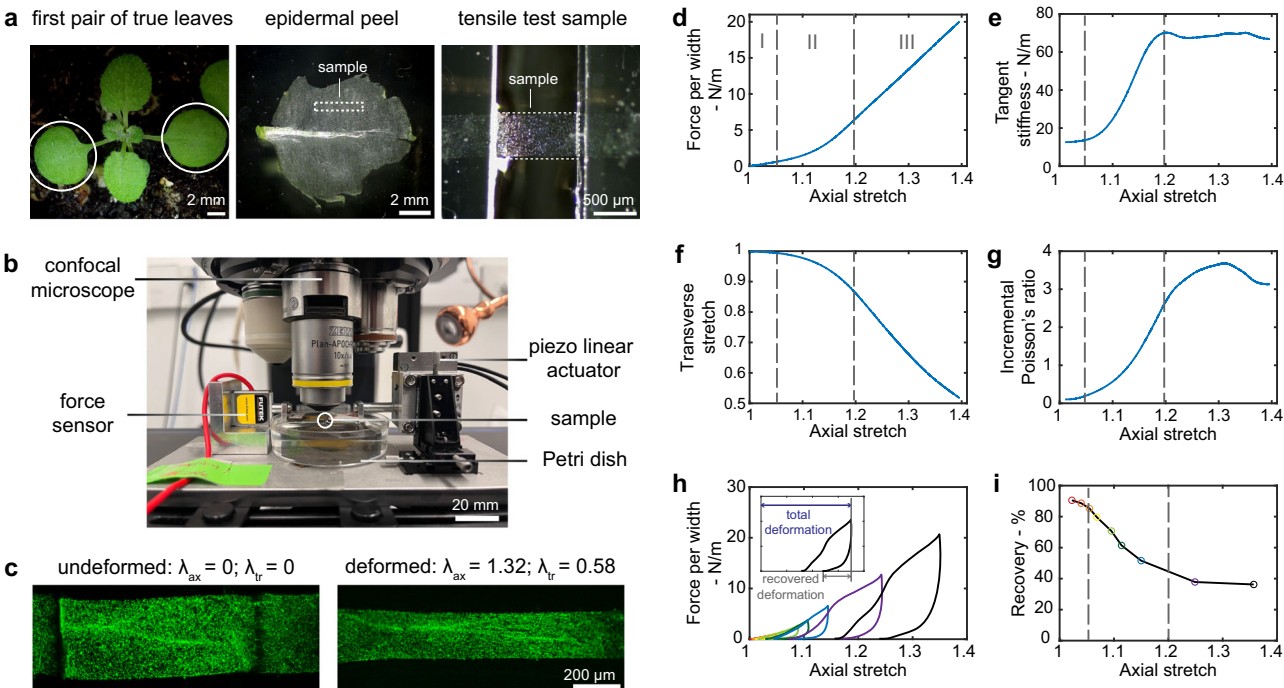

**Fig. 1 | Arabidopsis leaf epidermal peels have three-regime nonlinear mechanical behavior. a** Test samples from epidermal peels of the first pair of true leaves. **b** Micromechanical tensile stage coupling with confocal microscopy. **c** Confocal images of a sample at undeformed status (projection of a 3D stack) and deformed status (single optical section), $\lambda_{ax}$ as the axial stretch, $\lambda_{tr}$ as the transverse stretch. Representative **d** force per width - axial stretch curve, **e** tangent stiffness - axial stretch curve, **f** transverse - axial stretch curve, and **g** incremental Poisson's ratio - axial stretch curve from monotonic tensile tests. Representative **h** force per width-axial stretch curve and **i** recovery−axial stretch from incremental cyclic tensile tests with loading-unloading cycles are distinguished by different colors. The recovery percentage is the percentage of recovered deformation relative to the total deformation. Regimes I, II, and III are divided by the dashed lines.

to the loading. Finally, in regime III, the tissue is relatively stiff, with a nearly constant tangent stiffness (~ 70 N/m, which is equivalent to 70 MPa) (Fig. 1d−e). The stiffness range is similar to that reported in tensile tests on Arabidopsis hypocotyls[40]. Thus, the force and the deformation of the tissue do not have a simple linear relationship, which suggests a complex relation between turgor-driven growth and turgor pressure. This complexity cannot be simply represented by a single stiffness value derived from a limited range of deformation but rather should be characterized as a function that varies with deformation.

When a material is stretched, it will not only deform in the direction of applied force (in the axial direction) but also in the transverse direction (Poisson effect); the relation between the deformation in both directions gives insight into the microstructure. In addition, knowing the Poisson effect is important to relate mechanical response to multi-direction in-plane loading (e.g., equibiaxial), which is experienced by the cell wall of a growing cell since turgor pressure stretches the cell wall in all directions. To understand more about the Poisson effect of the epidermal peels, we looked at the deformation in both axial and transverse directions. We note that no obvious wrinkling was observed during testing (Supplementary Fig. 2), indicating that the deformation occurred primarily in-plane. The transverse stretch (deformed width of the sample divided by the initial width of the sample) decreases (i.e., contracts) in a nonlinear manner as the axial stretch increases, which also corresponds to the three regimes previously identified (Fig. 1f). To quantify the relation between the transverse stretch and axial stretch, we calculated the incremental Poisson's ratio (Methods, Post-experiment analysis). An incremental Poisson's ratio of 1 here implies the conservation of surface area such that the increase in length is directly compensated by a decrease in width. In regime I, the transverse contraction is limited. The incremental Poisson's ratio is smaller than 1 and increases weakly with

stretching, indicating that the area of the specimen is increasing. In regime II, the transverse contraction intensifies, and Poisson's ratio increases and eventually is more than 1, which means that there is more transverse contraction than axial extension so that the area of the specimen is decreasing. In regime III, the transverse contraction stabilizes, and Poisson's ratio increases but with a declining rate, reaches a peak, and then decreases as axial stretch increases further (Fig. 1f−g). This unusual nonlinear Poisson effect distinguishes epidermal tissue from classical soft materials, such as rubbers, which typically have an incremental Poisson's ratio of 0.5 throughout deformation. An important question raised here is what underlying structure within the epidermal peels contributes to a large change in Poisson's ratio, which we will address in the next section.

The growth of a cell is due to irreversible deformation of the cell wall. To understand the underlying mechanism of irreversible (plastic) deformation, it is important to look at how this deformation evolves during stretching. We explore irreversible deformation by incremental cyclic tensile tests, in which the sample was loaded and unloaded (stretched and unstretched) cyclically with progressively increasing load levels until it failed. In each load-unloading cycle, the axial stretch at zero force point of the unloading curve is always larger than the immediately prior loading curve, indicating that irreversible deformation accumulates during stretching (Fig. 1h). The reloading curve is above the unloading curve, showing hysteresis and indicating the sample exhibits some viscous behavior in addition to plasticity. When the applied force exceeds the maximum force previously applied in the incremental cyclic test, the force per width versus the axial stretch response (Fig. 1h) follows the response from the monotonic tensile tests (Fig. 1d). To quantify how much deformation is recoverable during stretching (the ability to return back to the original size), we calculated the recovery percentage (Fig. 1i). The recovery versus axial stretch curve also exhibits a three-regime behavior. Most of the

deformation (more than 80%) is recoverable in regime I, the recoverable deformation sharply decreases from 80% to 40% in regime II, and then eventually levels off in regime III. The transverse-axial stretch response under cyclic loading also largely follows the monotonic behavior (Supplementary Fig. 3a). Unloading happens with reduced transverse strain recovery, and unlike for the axial stress-strain response, there is relatively little hysteresis between unloading and reloading. The recovery of transverse stretch shows a similar magnitude and trend to that of axial stress (Supplementary Fig. 3b). This aligns with the widely held view that the cell wall becomes plastic (yields) when the stress (stretch) passes the yield point[10]. In addition, epidermal tissue becomes increasingly plastic with continuous stretching. This observation raises the question of what underlying mechanisms control the gradually changing plastic behavior of the epidermal tissue.

In summary, our comprehensive examination of the mechanical behavior of the epidermal peel reveals that it displays nonlinear strain-stiffening, nonlinear Poisson effect, and nonlinear plasticity, all with the same three distinct regimes. This suggests that a common underlying mechanism governs these properties, which we will explore further in the next section.

## Nonlinear mechanical behavior comes from the cell wall—a fibrous network material

The nonlinear mechanical behavior of epidermal peels can likely be attributed to a combination of two aspects: the cellular structure (shapes and arrangement of cells in the layer) and the material composition of the cell wall. We will first delve into whether and how the cellular structure influences the overall mechanical behavior of the epidermal peel. To explore the contribution of the cellular structure to the stiffness and Poisson effect, we utilized the finite element method to compare the mechanical responses of the cellular structure with the mechanical response of a single flat plate under stretch, representing just the intrinsic properties of the cell wall (Supplementary Movies 2–3). We constructed cellular structures of the epidermal cells of the first pair of true leaves and single plate in silico (Fig. 2a), as described in Methods, Finite element simulations (Supplementary Software 1). After peeling, the epidermal cells lose their turgor pressure (Supplementary Fig. 1b). We further verified this by performing tensile tests on the frozen and thawed treatment samples (Supplementary Methods 1.3), and we showed mechanical behavior similar to that of untreated samples (Supplementary Fig. 1c). Therefore, in our cellular structure model, the top and bottom periclinal cell walls were treated as flat plates. The anticlinal walls were extruded from the cell outline boundary by the thickness of the cell which is 10 microns (Fig. 2b). The same purely elastic mechanical properties are assigned to the material of the plate and cellular structure (Methods, Finite element simulations). Comparing the cellular model with the flat plate model, we find that the force per width—axial stretch curves are very similar for these two models (Fig. 2c). The two force-stretch curves have a maximum 8.3% deviation within the tested range of deformation, indicating that the geometry of the cellular structure is only a minor factor in the force deformation response. Further, the cellular structure is not responsible for the non-linear strain stiffening observed experimentally. Likewise, the axial-transverse stretch curves of the cellular structures are very close to that of a single plate with a maximum 0.4% deviation within the tested range of deformation (Fig. 2c), indicating that the cellular structure has almost no impact on the Poisson effect. Poisson's ratio is not affected because there is no component of this cellular structure contributing to the lateral contraction. Varying the cell size of the cellular structure, the thickness of the cell wall, and the material model of the cell wall results in this same conclusion that the cellular structure has little impact on overall mechanical behavior (Supplementary Fig. 4). In other words, the nonlinear mechanical

properties of the epidermal peels comes from the material properties of the periclinal cell wall.

Now the question is what underlying mechanisms control our observed nonlinear mechanical behavior of the cell wall. The cell wall consists of cellulose microfibrils embedded in a matrix of pectin and hemicellulose[10]. Cellulose microfibrils contribute to the mechanical behavior in a manner that is strongly influenced by their orientation[1]. We hypothesize that the alignment of cellulose microfibrils during stretching leads to the nonlinear mechanical behavior of the cell wall. To examine this idea, we used cross-polarization microscopy to test the birefringence of the undeformed and samples deformed by an intermediate stretch (Methods, Cross-polarization microscope imaging). Cellulose microfibrils exhibit inherent birefringence[58], so the overall birefringence observed in the cell wall can provide insights into the orientation of these cellulose microfibrils within the cell wall. The similarity in overall brightness of the undeformed sample when oriented at 0 degrees and at 45 degrees suggests that the unstretched sample does not show birefringence. This observation implies that the cellulose microfibrils are initially distributed evenly in all orientations on average at the tissue level. We note that the guard cells, in which the cellulose microfibrils are highly aligned, appear brighter (Fig. 2d). In addition, there is brightness fluctuation within one cell, which is due to subcellular regions of anisotropic cellulose alignment in wavy pavement cells[59]. In contrast, the deformed sample oriented at 45 degrees is brighter, whereas at 0 degree is not, indicating the stretched sample becomes birefringent. This change indicates that cellulose microfibrils orient towards the axial direction overall as a result of stretching (Fig. 2e). This behavior parallels with the reorientation of cellulose microfibrils during unidirectional cell growth, as described by multinet growth theory[60].

We first assess whether the alignment of microfibrils based on an affine deformation assumption can account for the mechanical behavior of the cell wall. We adapt a simple affine isotropic network model[35], in which all fibers are stretched and rotate such that local deformation is consistent with global deformation (Fig. 2f; Supplementary Notes 1; Supplementary Software 2). The force per width, tangent stiffness, transverse stretch, and incremental Poisson's ratio versus the axial stretch curves resulting from this affine assumption (Fig. 2g) all show different trends compared to the experimental results (see previous section). This model predicts that the strain stiffening behavior has only two regimes instead of three regimes: stiffness gradually increases in regime I and levels out in regime II. In addition, this model predicts that the incremental Poisson's ratio increases monotonically instead of increasing until reaching a peak and then decreasing, as in the experimental data. Therefore, the nonlinear mechanical behavior of cell walls under stretch cannot be solely attributed to the reorientation of cellulose microfibrils resulting from affine deformation.

Two key aspects overlooked in the aforementioned affine model are that cellulose microfibrils can bend/unbend[61] in addition to stretch/unstretch (i.e., beam-like in mechanics terminology), and that these microfibrils can slide relative to one another, deforming and sliding through the amorphous matrix[1]. Because of their large length-to-diameter aspect ratio, cellulose microfibrils close to the transverse direction tend to bend rather than shorten in length axially, which reduces the resistance to transverse deformation. To incorporate this bending under compression, we adapted a five-beam model[33,62] utilizing a simple diamond structure of five beams (Fig. 2h; Supplementary Notes 2; Supplementary Software 2), which we use to conceptualize the arrangement structure of cellulose microfibrils in the cell wall. In this model, beam AB represents cellulose microfibrils close to the transverse direction; it undergoes bending if the structure is stretched along the indicated force direction. Beams AC, AD, BC, and BD then represent the majority of cellulose microfibrils, which rotate towards the direction of applied force and elongate elastically. The

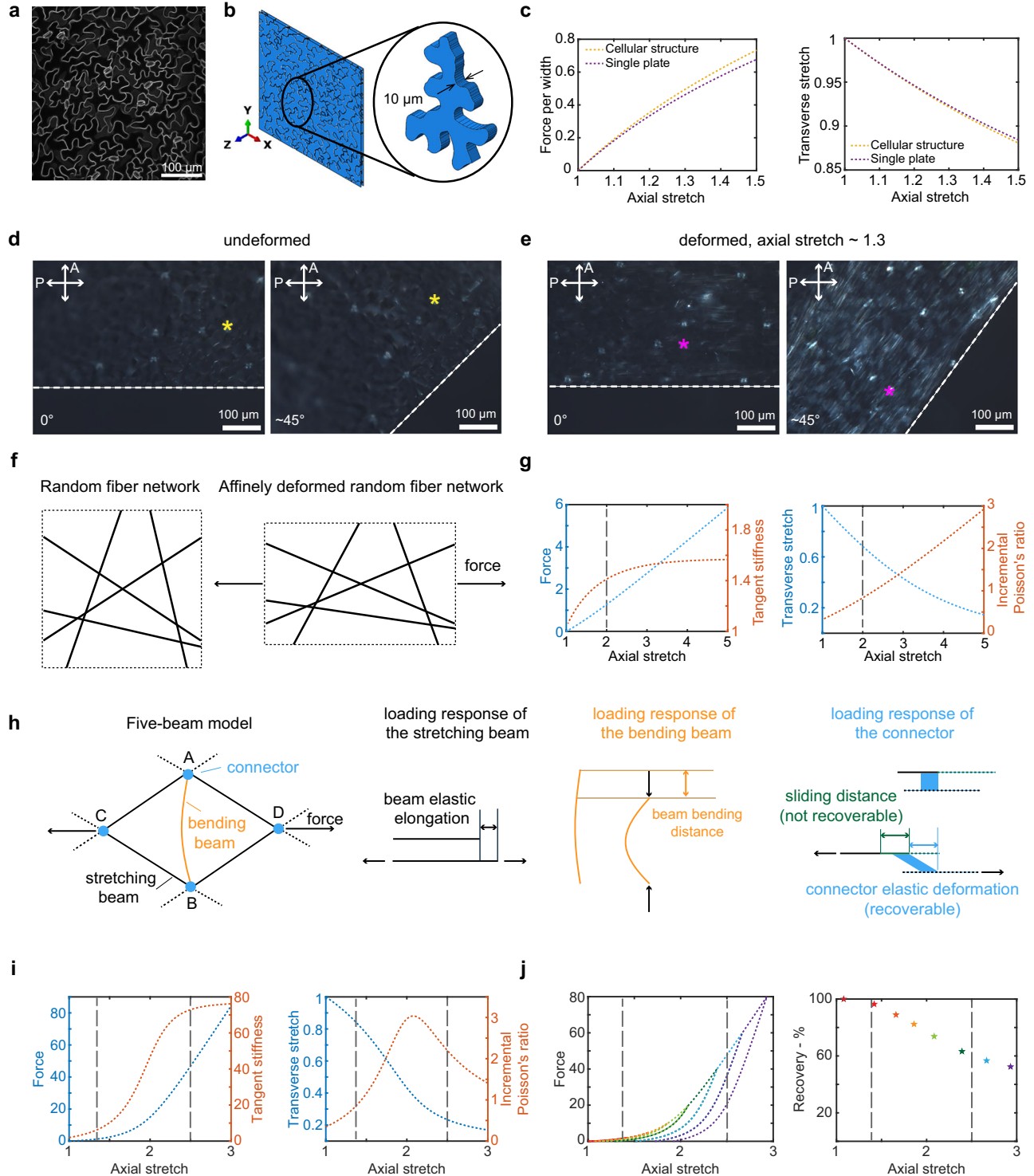

**Fig. 2 | Physical mechanisms that could be underlying the stress response of the epidermal peel to deformation. a** Maximum projection of the confocal z-stack image of leaf abaxial side epidermal cells with the membrane marker, and based on it **b** the cellular structure of the epidermis is constructed in Abaqus. The inset shows an isolated cell, with two arrows indicating the thickness of the cell. **c** Force per width versus axial stretch curves and Transverse stretch versus axial stretch curves of FEM simulation results of three cellular structures and a single plate. Cross-polarized microscope images at 0 and 45 degrees between the axial direction and polarizer of **d** an undeformed sample and **e** a deformed sample. The star marker indicates the same point of the sample oriented at different angles. The experiment

was repeated three times with similar results. **f** Schematic of the affine isotropic network model at undeformed and affine deformed status respectively. **g** Force (left axis) and tangent stiffness (right axis), transverse stretch (left axis), and incremental Poisson's ratio (right axis) versus axial stretch curves of the affinely isotropic network model. **h** Schematic of the five-beam conceptual model and loading responses of stretching beam, bending beam and connector respectively. **i** Force (left axis) and tangent stiffness (right axis), transverse stretch (left axis) and incremental Poisson's ratio (right axis) versus axial stretch curves of monotonic loading, **j** force and recovery percentage versus axial stretch curves of loading-unloading of the five-beam conceptual model.

connector in our model, which we have added to the original five-beam concept, represents the links between cellulose microfibrils. These links transfer mechanical forces from one microfibril to another. Their physical nature could result from direct cellulose-cellulose contact, interactions mediated by the polysaccharide matrix, or both[10,63]. To capture the relative sliding of microfibrils within the cell wall causing plastic behavior[1], we assume that if the force exceeds a certain threshold, the connector will not only deform but also slide between beams. Note that we are not comparing the exact values, as our model uses a simple geometry while the actual structure of cellulose microfibrils in the cell wall is much more complex - we aim to represent qualitative trends.

The force per width, tangent stiffness, transverse stretch, and incremental Poisson's ratio versus the axial stretch curves for the five-beam model (Fig. 2i) all show similar trends compared to experimental results (see previous section), exhibiting nonlinear mechanical behavior with three regimes. The recovery behavior of the model (Fig. 2j) also shows a similar trend to the experiments, where the recovery gradually decreases with increased stretch (see previous section). From this five-beam model, the three characteristics of the cell wall's nonlinear mechanical behavior can be understood to result from the transition of the deformation mode of cellulose microfibrils from reorientation and bending-dominated to stretch-dominated. (1) Strain-stiffening behavior: the microfibrils close to the transverse direction tend to bend since it is energetically favorable initially, a small amount of force can deform the network, so the stiffness of the network is low; as the network deforms, the microfibrils close to the axial direction align and a larger force is needed to deform the network, so the stiffness of the network is gradually increasing; eventually, the angle of the side microfibrils stabilize so axial deformation dominates, and the stiffness of the structure reaches a plateau. (2) Poisson effect: as the microfibrils close to the transverse direction bend, it leads to more transverse contraction than if it were compressing axially, causing an increase in Poisson's ratio. As the network deforms, it becomes harder to further bend the transverse microfibrils, which leads to a decrease in the incremental Poisson's ratio. (3) Irreversible deformation: when the bending mode dominates, the transmitted force between the microfibrils is relatively small resulting in minimal fiber-fiber sliding. As fiber stretching mode starts to dominate, the transmitted force gradually increases, resulting in more sliding, thus irreversible deformation. Eventually, recovery reaches a plateau, suggesting fiber stretching mode dominates and the deformation contributions from fiber-fiber slippage and elasticity have stabilized. This implies that the transition of actions of cellulose microfibrils from reorientation and bending-dominated to stretch-dominated leads to the nonlinear stiffening and nonlinear Poisson effect of the cell wall, and also a gradual change in plasticity.

## Mechanical properties of the cell wall change during development

During development, the correct formation of plant organs is ensured by properly coordinated growth rates, which are influenced by the mechanical properties of the cell wall. In this section, we will examine changes in the overall mechanical behavior of the cell wall during development and how these changes relate to growth rates. We will also discuss the possible mechanisms that control these alterations in mechanical behavior.

To investigate the mechanical behavior of the cell wall at various developmental stages with distinct growth rates, we conducted monotonic tensile tests on the epidermis of the first pair of true leaves at 12-day, 18-day or 25-day post-germination, corresponding to relatively fast growth, slow growth, and no growth respectively (Fig. 3a). For reference, measurements in previous section were made at 25 days. Leaf epidermises from all developmental stages show three-regime strain-stiffening behavior as discussed before (Fig. 3b). The

stiffness $E_1$ in regime I (the slope of the initial linear regime) of samples at different stages are not significantly different, while the stiffness $E_2$ in regime III (the slope of the final linear regime) increases with high statistical significance as the samples get older. Therefore, growth rate is associated with the stiffness $E_2$ but not $E_1$, and a slower growth rate corresponds to a higher $E_2$. This result aligns with the widely held view that stiffness serves as an indicator of wall extensibility, which in turn controls the growth rate[64]. Moreover, it is important to consider stiffness beyond regime I, as stiffness in this initial regime tends to remain consistent throughout development.

To determine how Poisson's ratio changes over development, which provides insights into deformation mechanisms, we examined the transverse stretch versus axial stretch curves and found they are similar among different stages (Fig. 3c). Additionally, there are no statistically significant differences in the initial incremental Poisson's ratio $v_1$ and the peak Poisson's ratio $v_2$ among these stages (Fig. 3c).

To explore the plastic behavior at different developmental stages, we conducted incremental cyclic tensile tests on leaf epidermal peels. The plastic behavior of samples from various developmental stages consistently shows three regimes: most deformation is recoverable in regime I, recovery decreases sharply in regime II, and levels off in regime III (Fig. 3d). At the same stretch level, the recovery percentage of deformation is similar across different development stages. Interestingly, the force per width versus residual stretch curves vary across these three stages: at the same load level the accumulated residual stretch is larger in earlier stages than in late stages. This result implies that older cells tend to accumulate less plastic deformation than younger cells at the same level of turgor pressure.

To explore the possible mechanisms behind the variation in the stiffening behavior of cell walls at different stages, we revisit the five-beam model in the previous section. As we found previously, the mechanical properties of the epidermis come from the cell wall materials, whereas the cellular structures at these different developmental stages have little influence on the overall mechanical behavior (Supplementary Fig. 4b–c). For fibrous network materials, factors such as fiber density, fiber orientation, and crosslinks between fibers can significantly influence the stiffness[33]. We first consider the increase in fiber numbers due to more cellulose microfibrils being deposited into the cell wall as it grows. This, however, would scale $E_1$ and $E_2$ equally, contradicting the experimental results. Therefore, more deposited cellulose microfibrils cannot explain the stiffness variation. Additionally, more alignment of cellulose microfibrils in the proximal-distal directions during development is unlikely, since the mechanical behavior in both medial-lateral and proximal-distal directions are quite similar at all development stages (Supplementary Fig. 5). We then examine the impact of crosslinks by varying the connector deformation resistance of the five-beam model. The results show that the connector deformation resistance does not influence the initial slope of the force-stretch curve but significantly influences the later stiffening behavior (Fig. 3e). A higher connector deformation resistance results in a high final stiffness $E_2$ but maintains a similar initial stiffness $E_1$. This is because the initial stiffness is dominated by the bending and orientation of the fibers, while the final stiffness is dominated by the fiber stretch and slip. This suggests that the variations in stiffness at different stages may be primarily due to changes in the connections between cellulose microfibrils.

The changes in the deformation resistance of the connector can also explain some of the trends in the Poisson effect and plastic behavior of the cell walls at different stages. The changes in the deformation resistance of the connector do not affect the initial Poisson's ratio, aligning with experimental observations (Fig. 3f). A higher resistance leads to an increased peak Poisson's ratio in the model, whereas experiments show this trend but without a significant difference. This discrepancy may be attributed to the simplified

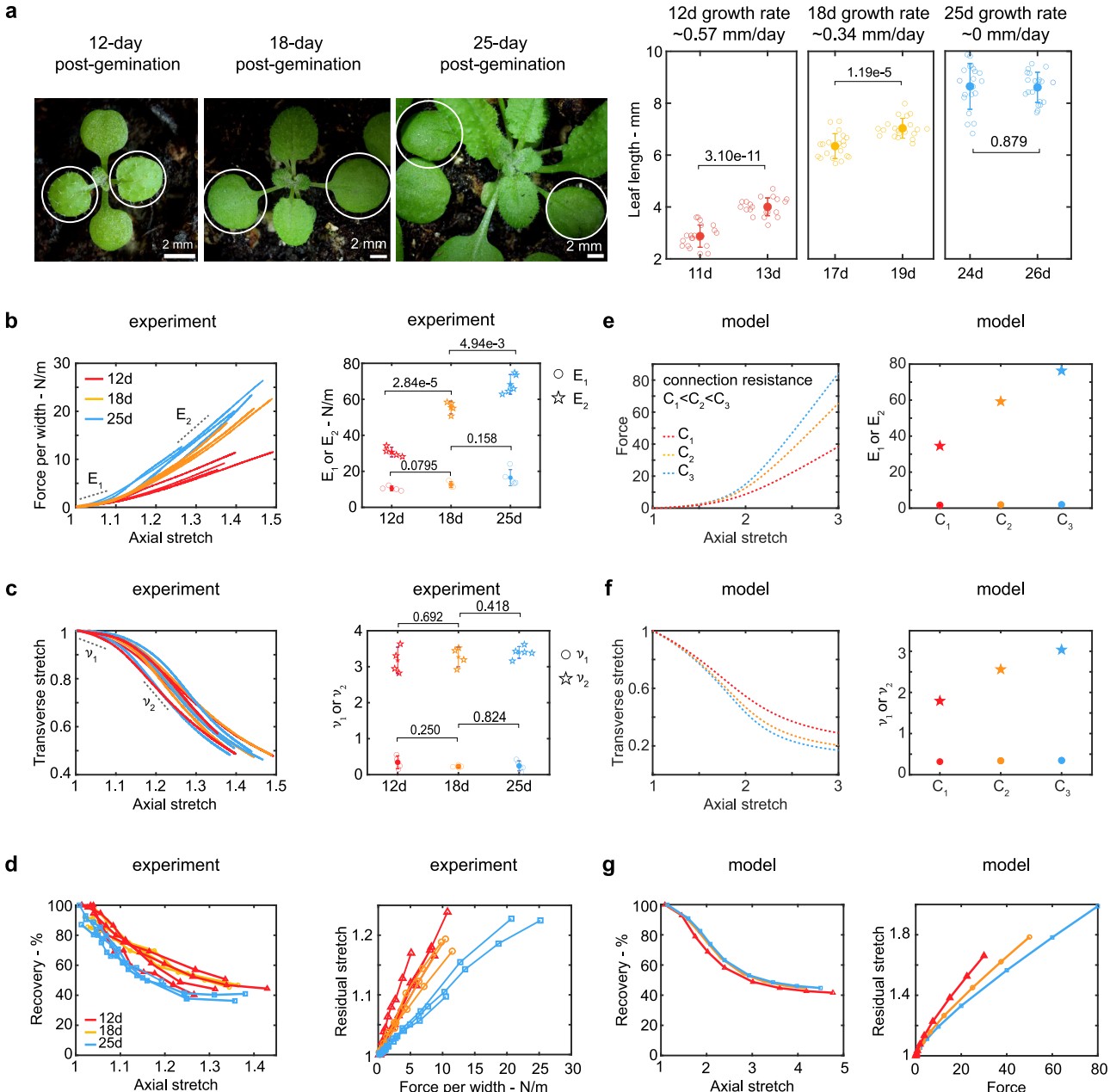

**Fig. 3 | Changes in the mechanical properties of leaf epidermis across developmental stages and potential underlying mechanisms. a** First two true leaves of plants for three developmental stages (12, 18, and 25 days pot-germination) and their growth rates estimated based on leaf length measurements from one group of plants one day before, and from a different group of plants one day after each stage ($n = 20$ for each group; samples from twenty individual plants as biological replicates). Individual data points are shown in hollow markers. The solid markers represent the means. The error bars represent the means ± SD. Two-tailed unpaired Student's $t$-test. **b** Mechanical force - axial stretch curves and scatter plot of the initial ($E_1$) and the final stiffness ($E_2$), **c** transverse - axial stretch curves, and scatter plot of the initial ($v_1$) and peak Poisson's ratio ($v_2$) for 12-day ($n = 4$; samples from four individual plants as biological replicates), 18-day ($n = 4$; samples from four individual plants as biological replicates), and 25-day ($n = 5$; samples from five individual plants as biological replicates) stage from monotonic tensile tests. Note that one data from the 25-day stage is reused from Fig. 1d–f. Individual data points are shown in hollow markers. The solid markers represent the means. The error bars represent the means ± SD. Two-tailed unpaired Student's $t$-test. **d** recovery percentage versus axial stretch curves and residual stretch versus membrane stress for 12-day ($n = 4$; samples from four individual plants as biological replicates), 18-day ($n = 3$; samples from three individual plants as biological replicates), and 25-day ($n = 3$; samples from three individual plants as biological replicates) stage from incremental cyclic tests. Note that one data from the 25-day stage is reused from Fig. 1i. **e** Force per width - axial stretch curves, and plot for the initial ($E_1$) and final stiffness ($E_2$), **f** transverse stretch - axial stretch curves, and plot for the initial ($v_1$) and peak Poisson's ratio ($v_2$), and **g** recovery percentage - axial stretch curves from the five-beam model by varying the deformation resistance of connectors.

structure of the five-beam model, which does not fully represent the complexity of cellulose microfibrils in cell walls.

Exploring the same five-beam model for plastic behavior (Fig. 3g), we found that the model accurately captures the similarity in recovery-stretch curves across different development stages. The model also shows that the residual stretch is smaller at the same force level for

older development stages. This suggests that plastic behavior is dominated by stretch rather than force.

## The anisotropy of the cell wall is affected by microtubules

During plant development, the regulation of anisotropic growth generates various organ shapes. It is generally accepted that this

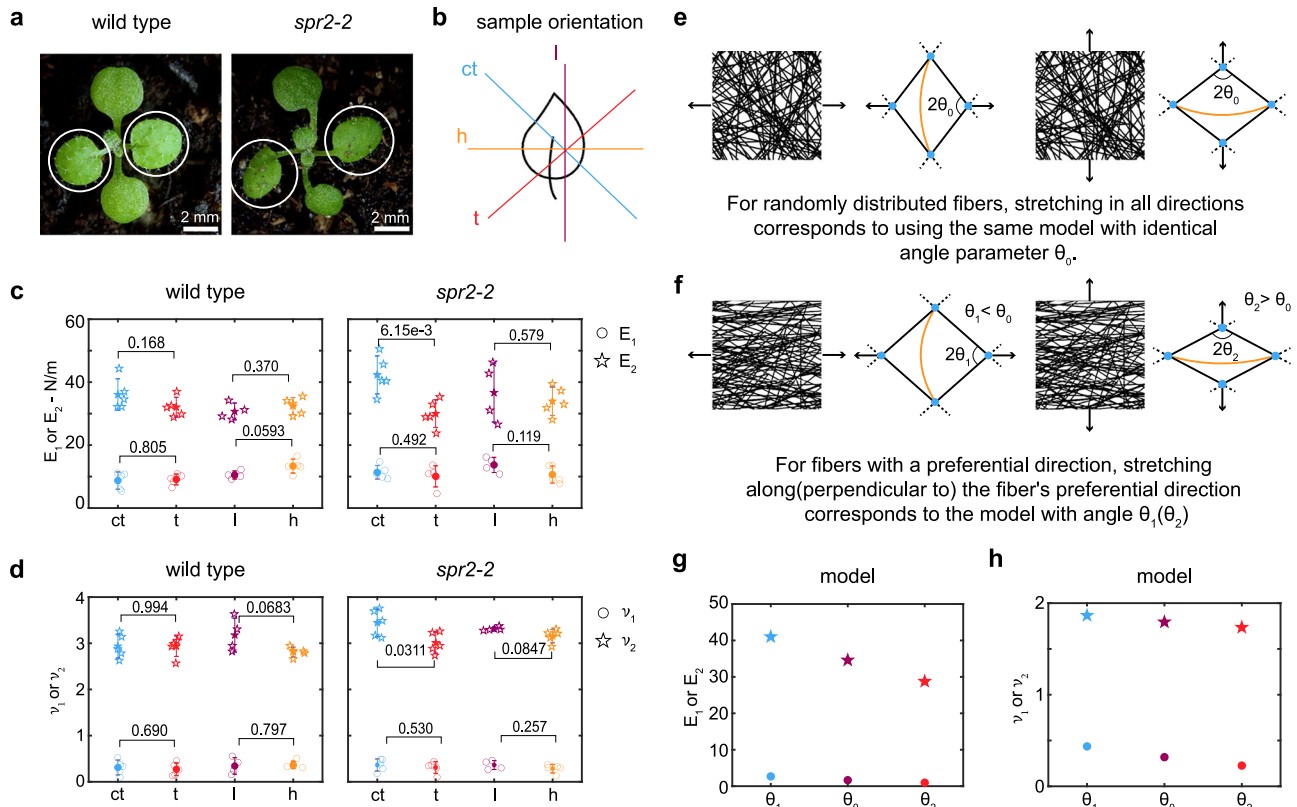

**Fig. 4 | *spr2-2* mutant shows anisotropic mechanical properties, implying enhanced alignment of cellulose microfibril. a** first two true leaves of wild type and *spr2-2* mutant. **b** Preparation of the samples from the leaf epidermis in four orientations (l, h, ct, and t). In the schematics, the peeled abaxial epidermis is in the normal orientation (cuticle side oriented away from the view). Scatter plot displaying **c** the initial ($E_1$) and the final stiffness ($E_2$) **d** the initial Poisson's ratio ($v_1$) and the peak Poisson's ratio ($v_2$) for four orientations of wild-type and *spr2-2* mutant samples. For the wild type: ct ($n = 5$; samples from five individual plants as biological replicates), t ($n = 5$; samples from five individual plants as biological replicates), l ($n = 4$; samples from four individual plants as biological replicates) and h ($n = 5$; samples from five individual plants as biological replicates). For *spr2-2* mutant: ct ($n = 5$; samples from five individual plants as biological replicates), t ($n = 5$; samples

from five individual plants as biological replicates), l ($n = 4$; samples from four individual plants as biological replicates), and h ($n = 5$; samples from five individual plants as biological replicates). Individual data points are shown in hollow markers. The solid markers represent the means. The error bars represent the means ± SD. Two-tailed unpaired Student's t-test. **e** Schematic of the five-beam model representing randomly distributed fibers under loading in both horizontal and vertical directions. **f** Schematic of the five-beam model with varying angles between two stretching beams, representing fibers with a horizontal preferential orientation under loading in horizontal and vertical directions. Scatter plot of **g** the initial ($E_1$) and the final stiffness ($E_2$), **h** the initial Poisson's ratio ($v_1$) and the peak Poisson's ratio ($v_2$) from the model with two different angles.

anisotropy is generated by the anisotropic mechanical properties of the cell wall[8]. To investigate this relationship, we compared cell wall anisotropy between wild-type plants and a mutant with altered leaf shape. Given that microtubules guide the deposition of cellulose microfibrils in the cell wall and are closely associated with anisotropic mechanical properties[12], we selected a microtubule-related mutant, *spiral2* (*spr2-2*). A *spr2-2* mutant plant does not have functional SPR2 protein, which is a microtubule-associated protein that binds to the minus end of microtubules and stabilizes them[65–67]. As a result, microtubules in this mutant are more dynamic which leads to more aligned/anisotropic microtubules in sepals of *spr2-2* than in wild-type Arabidopsis[68]. In wild-type plants, the first pair of true leaves grow isotropically and form a flat circular blade (Fig. 4a). However, in the *spr2-2* mutant, the first pair of true leaves are twisted and curl in a counterclockwise orientation (Fig. 4a), implying altered cell wall anisotropy in this mutant. To test this hypothesis, we compare the cell wall anisotropy between the *spr2-2* mutant and wild-type plants and explore the possible mechanisms underlying alterations in mechanical behavior.

To investigate the difference in anisotropic mechanical behavior of the cell wall between *spr2-2* mutant and wild type, we conducted monotonic tensile tests on the epidermis of the first pair of true leaves at 12 days post-germination when the curled leaf phenotype begins to

appear. We measured mechanical properties in four orientations: counterclockwise tilted 45 degrees relative to the midrib (ct), clockwise tilted 45 degrees relative to the midrib (t), proximal-distal (l), and medial-lateral (h) (Fig. 4b). These orientations were chosen to provide a comprehensive assessment of anisotropy, examining mechanical behavior in axial, transverse and both diagonal directions relative to the main midrib axis of the leaf. Results of monotonic tensile tests show that samples from all four directions of both the mutant and wild type exhibit the same three-regime behavior (Supplementary Fig. 6). For the *spr2-2* mutant, the initial stiffness $E_1$ and initial Poisson's ratio $v_1$ show no significant difference between ct and t or between l and h orientations. However, the final stiffness $E_2$ (the slope of the final linear regime) and peak Poisson's ratio $v_2$ in the ct orientation are statistically significantly higher than in the t orientation, while in l and h orientation have no significant difference (Fig. 4c–d). For wild type, there are no significant differences in the initial stiffness $E_1$, final stiffness $E_2$, initial Poisson's ratio $v_1$ and peak Poisson's ratio $v_2$ between ct and t orientations, or l and h orientations (Fig. 4c–d). This indicates mechanical properties are anisotropic in *spr2-2* mutant, in contrast to the wild type, which are isotropic. This anisotropy of *spr2-2* likely originates from a slightly enhanced fiber alignment in the ct direction (given that $E_2$ in this direction is 40% greater than the t-direction) and likely contributes to leaf curvature.

The five-beam model offers insights into anisotropic mechanical behavior. To determine if fiber alignment can cause these anisotropic behaviors, we utilized the model again. For isotropic mechanical behavior, we used the same parameters with the angle parameter ($\theta_0$) of the model as originally employed (Supplementary Notes 2) in all directions to produce the same mechanical behavior (Fig. 4e), setting it as the baseline. To examine how the preferential cellulose microfibrils orientation influences mechanical properties, we alter the angle $\theta$ between two stretching beams in the five-beam model (Fig. 4f). We used a smaller angle parameter ($\theta_1 < \theta_0$) to represent the preferential cellulose microfibrils orientation along the stretching direction, and a larger angle parameter ($\theta_2 > \theta_0$) to represent the preferential cellulose microfibrils orientation perpendicular to the stretching direction. We found that the $\theta_1$ case produces a similar initial stiffness $E_1$ but a significantly higher final stiffness $E_2$ than $\theta_0$ case, whereas the $\theta_2$ case produces a lower final stiffness $E_2$ than $\theta_0$ case (Fig. 4g). Additionally, the $\theta_1$ small angle case produces higher $v_1$ and $v_2$ whereas $\theta_2$ large angle case produces lower $v_1$ and $v_2$ compared to $\theta_0$ case (Fig. 4h). This indicates that the enhanced alignment of cellulose microfibrils in the cell wall of *spr2-2* is close to the ct direction, or in other words, nearly perpendicular to t direction, leading to observed anisotropic behaviors. We note that $v_1$ is similar in all orientations in the experiments, which might be attributed to the simplicity of the five-beam model enhancing the sensitivity of $v_1$ to the initial geometry.

## Discussion

In this study, we investigated the mechanical behavior of the Arabidopsis leaf epidermis. We explored how mechanical properties change across a large deformation regime over different developmental stages. Our findings reveal that: (1) the leaf epidermis exhibits distinct three-regime nonlinear mechanical behavior, arising from the intrinsic deformation mechanisms of the cell walls, which are fibrous network materials; (2) mechanical properties vary across developmental stages associated with different growth rates, potentially due to changes in the connections between cellulose microfibrils; (3) the *spr2-2* mutant with curled leaves shows anisotropic mechanical properties, likely linked to the enhanced alignment of cellulose microfibrils.

In our study of peeled Arabidopsis leaf epidermis, the cellular structure has limited influence on the tensile mechanical behavior at the tissue level, as indicated by our finite element simulation. This is because pavement cells are deflated and have very wavy anticlinal walls, which offer minimal mechanical resistance during stretching. In many eudicot plants, pavement cells have this jigsaw puzzle shape[8], suggesting that this finding could also apply to these cases. However, this conclusion may not extend to tissues composed of polygonal cells, such as the onion epidermis, where the anticlinal wall could be reoriented and stretched when the tissue is deformed[69,70]. In such cases, the cellular structure is more likely to influence overall tissue mechanics. In addition, we note that subcellular details are simplified in our FEM model, where we assume uniform thickness and homogeneous mechanical properties for both periclinal and anticlinal walls. Although we believe these simplifications do not substantially impact our main conclusions, they should be reconsidered if precise deformation patterns at the subcellular scale are of particular interest.

The observed three-regime nonlinear strain-stiffening behavior, known as a J-shaped stress-strain curve, is typical of many soft mammalian biomaterials[71]. This behavior allows animal tissues to undergo deformations within a range while preventing excessive deformation that could lead to damage when stress exceeds this range[72], and likely provides similar benefits for plant tissues. According to Laplace's law, slight turgor pressure changes can cause large deformations of plant cells if considering linear elasticity, but stiffening behavior could prevent this (Supplementary Notes 3). This nonlinear stiffening has also been observed in other plant tissue, like apple fruit and potato tuber tissues, though it is unclear whether it comes from the tissue structure

or individual cells, and the underlying mechanisms have not been extensively studied[73,74]. In contrast, the primary cell wall of the onion bulb, often examined for its mechanical properties, exhibits a different pattern where stiffening is typically followed by softening[47,54]. These differences highlight that the mechanical behavior of cell walls can vary between different organs, species, and growth stages. Additionally, it has been proposed that strain-stiffening behaviors can vary within different regions of the same organ, influencing growth patterns and organogenesis[75]. The differences in mechanical behavior are worth further exploration.

There is less literature on Poisson's ratio of plant tissues compared to stiffness measurements, and it is often treated as a constant, like 0.5, in interpreting indentation tests[41]. To our knowledge, our work is the first to demonstrate how Poisson's ratio changes with stretch across a large deformation regime. Understanding Poisson's ratio is crucial not only for providing insight into deformation mechanisms, but also for analyzing the deformation of plant cells or tissues under complex loading conditions (Supplementary Notes 4). For instance, in a pressurized cylindrical cell stretched along its long axis, knowing Poisson's ratio is essential for calculating the cell's overall stiffness. A higher Poisson's ratio in the cell wall would make the cell more resistant to deformation (Supplementary Notes 4.1, Case II). If Poisson's ratio increases with deformation, as observed in regime I, the cell becomes progressively more resistant, thereby reducing the impact of forces that come from neighboring cells or the environment. These concepts are extensible to large deformation, although more information is needed to draw direct conclusions (Supplementary Notes 4.2).

The plastic behavior observed here in Arabidopsis is also present similarly in onion bulb epidermal cell walls, where it results from the sliding of cellulose microfibrils[47]. Notably, while this sliding leads to greater plastic deformation as the cell wall deforms, it does not necessarily weaken the cell wall in an Arabidopsis leaf. The strain-stiffening beyond the previous maximum load level is similar to that observed in monotonic tests. This combination of strain-stiffening with plastic deformation is known as hyperelastic-plastic behavior[37] and has been observed in physically crosslinked nanocellulose hydrogels[76], where it is explained by the breaking and reforming of physical crosslinks. This suggests that during sliding, the connections between cellulose microfibrils may temporarily disassociate and reassociate, maintaining the integrity of the cell wall.

The observed trends in these three mechanical behaviors suggest that the cell wall behaves like a fibrous network material. Fibrous network materials are characterized by fibers forming an interconnected network, with or without a matrix, where the overall behaviors emerge from the collective deformation of the fibers and their interactions[33]. This is distinct from fiber-reinforced composites, where mechanical properties result from the additive contributions of both the matrix and the fibers[37,77]. Fibrous network materials encompass a broad range of materials, including nonwovens[78], mycelium[79], blood clots[80] and mammalian skin[81], all of which exhibit diverse mechanical behaviors. Notably, mammalian skin, a well-studied living material, shares significant similarities with plant epidermal cell walls. Skins also have stiff fibers - collagen fibers - as their primary load-bearing elements, forming a complex three-dimensional network[82]. Skin also exhibits a J-shape strain-stiffening behavior and a large Poisson effect when subjected to axial stretch[83], which comes from the collective behavior of straightening, aligning, and stretching of collagen fibers[84]. This suggests a similar deformation mechanism happening in the cell walls. However, there are two key differences between mammalian skin and cell walls: (1) collagen fibers are more flexible than cellulose microfibrils, resulting in a more pronounced tortuosity in the collagen network; (2) sliding occurs between cellulose microfibrils, while collagen fibers have less sliding motion[82]. These differences lead to slightly different mechanical behavior of skin

compared to plant cell walls: (1) strain-stiffening is more pronounced in skin because collagen fiber straightening requires less force in the first regime, and the covalent crosslinks preventing sliding allows greater stiffening in the third regime; (2) the deformation in skin is mostly recoverable due to the covalent crosslinks between collagen fibers. Given the well-established research on skin mechanics, comparisons between the two systems may offer valuable insights that could enhance our understanding of cell wall mechanics, which is worth exploring in future research.

We conceptualized the arrangement of cellulose microfibrils as a simple diamond-shaped structure and force is transmitted through connectors in the five-beam model. The physical nature of the connectors may arise from the polysaccharide matrix and direct cellulose-cellulose contacts. Xyloglucan, the dominant hemicellulose in the cell wall, can bind to cellulose microfibrils by hydrogen bonds or be trapped in the microfibrils[85]. Although evidence suggests that the majority of xyloglucan does not contribute to wall mechanics, a small fraction might intertwine with cellulose microfibrils to form junctions controlling wall mechanics, known as the "biochemical hotspots" hypothesis[22]. In addition, pectin with neutral sugar side chains can bind cellulose, likely through hydrogen bonds, potentially providing linkages[86,87]. However, it remains unclear how strong this interaction is[88]. Direct cellulose-cellulose contacts have recently received more attention, suggesting they might be the strongest connections transferring tensile forces within the cellulose network[32,89]. This model captures the key deformation modes of cellulose microfibrils: bending, reorientation, stretching, and slipping. It demonstrates that the observed three-regime nonlinear mechanical behavior arises from a transition between bending- and reorientation-dominated modes to stretching- and slipping-dominated modes. However, we note that the actual structure of the cell wall is far more complex since it is irregular and multilayered. If we imagine the wall as a distribution of diamond-shaped units with different sizes, each unit may undergo its bending-to-stretching transition at a different strain level. These local transitions may not only occur asynchronously but could also influence each other through network connectivity. In future work, we plan to incorporate more realistic cell wall architectures into our model to quantitatively predict mechanical responses.

The fibrous network structure of cell walls enables a high degree of tunability in their mechanical properties, allowing cells to adjust these properties to support the proper formation of plant organs. Our results indicate that as the leaf develops, the stiffness in the third regime increases, and the plastic deformation at the same stress level decreases. These changes are likely due to alterations in the connections between cellulose microfibrils. Additionally, the orientation of cellulose microfibrils is another critical factor influencing mechanical properties. Our findings indicate that the *spr2-2* mutant exhibits anisotropic mechanical behavior, likely due to the enhanced alignment of cellulose microfibrils, while the wild type displays isotropic behavior. We believe that viewing the cell wall as a fibrous material, where mechanical behavior arises from the collective motion of cellulose microfibrils and exhibits a high degree of tunability, can be extended beyond the specific leaf ages and positions within plants examined in this study. This perspective may apply more broadly to primary cell walls, as their main structures are similar to each other and consist of cellulose microfibrils embedded in a pectin and hemicellulose matrix. However, further work is needed to confirm its applicability across a broader range of developmental stages, leaf positions within plants, organs, and plant species.

There are several possible physical mechanisms causing alterations in the connections between cellulose microfibrils during plant development. One possibility is modification of the polysaccharide matrix. For example, changing the concentration of hemicellulose influences the tensile mechanical properties of a cell-wall analog material[90,91]. During growth, newly secreted matrix polysaccharides, such as pectin and hemicellulose, are integrated into the existing cellulose network by enzymatic and spontaneous crosslinking mechanisms[92], and such crosslinking may enhance physical connections between cellulose microfibrils. As cells mature, homogalacturonan (HG), a major pectin component, becomes less methyl-esterified, allowing more calcium ionic crosslinking and resulting in stiffer pectin gels[93]. To assess whether reduced HG content affects mechanical behavior, we performed tensile tests on *qua2-1* (Supplementary Methods 1.1), a mutant with around 50% less HG and defective cell adhesion[94,95]. We found that the initial stiffness (stretch from 1 to 1.05) and the early strain-stiffening behavior (stretch from 1.05 to 1.1) were comparable to the wild type. However, the stiffening regime ended earlier due to softening and rupture (Supplementary Fig. 7). This suggests HG reduction does not significantly affect cellulose microfibril connections during early deformation but may compromise the strength of these connections and cell adhesion. Additionally, direct cellulose-cellulose interactions may play a more critical role in altering connections between cellulose microfibrils compared to matrix polysaccharides. A coarse-grained physical model of epidermal cell walls[32] shows that strengthening cellulose-cellulose contacts increases wall stiffness and decreases plasticity while increasing the binding energies between cellulose and matrix polysaccharides has little effect. These cellulose-cellulose contacts can be influenced by factors such as water and xyloglucan[18].

Overall, our study highlights the intricate mechanical behavior of the Arabidopsis leaf epidermal cell walls and underscores their fibrous network nature. This understanding could lead to new insights into the relationship between mechanical properties and plant morphogenesis.

## Methods

### Plant materials
*Arabidopsis thaliana* accession Col-0 plants were used as wild type, and the *spr2-2* (AT4G27060) mutant is in the Col-0 background[66,68]. Seeds were sown on Lambert Mix LM-111 soil and then kept at 4 °C for two days. After stratification, seeds were transferred to a growth chamber; this event defines post-germination day 0. Plants were grown under long-day light conditions (16 h light/8 h dark). Temperature and humidity conditions in the growth chamber were maintained at 22 °C and 50% RH, respectively.

### Leaf epidermal peel and tensile test sample preparation
The first pairs of true leaves were dissected from the plants. We used custom gelatin-coated cloth (Supplementary Methods 1.3) to separate the abaxial epidermal peel from the leaf (Supplementary Fig. 8). We note that it is consistent that abaxial epidermis is removed from the leaf. Thin rectangular strips approximately 0.4 mm wide were cut from the epidermal peels at the desired orientation within the leaf. Each sample was immersed in a fluorescent bead suspension (Magsphere AMF-300, diluted 2.5 μL into 250 μL water) for 15 min, and then gently swirled in water to remove excess fluorescent beads before testing. The strips were then gripped at both ends using Gorilla Crystal Clear Tape, leaving an ungripped length (avoiding the midrib) of approximately 0.9 mm. Two strips of aluminum foil were used to prevent the sample from stretching (Supplementary Fig. 9A). Then, the sample with the gripped tape was attached to the custom micromechanical tensile stage using magnets (Supplementary Fig. 9B–C). The Petri dish was filled with water, and the sample was placed on the water surface to maintain moisture and flatness. The aluminum foil strips were cut before testing.

### Tensile test procedure
For the monotonic tensile test, the sample was stretched at a constant velocity of approximately 2.5 μm/s using the piezo linear stage until failure. The force exerted on the sample was monitored continuously by the load cell. Simultaneously, the pattern of fluorescent beads on the sample was captured continuously by a confocal microscope (Zeiss

710 upright laser scanning confocal microscope with a 10x lens) as time-series images at 1s intervals. During testing, the confocal stage was manually adjusted to ensure the specimen remained in the center of the imaging window. For the incremental cyclic tensile test, the sample was initially stretched at a constant velocity of approximately 2.5 μm/s and then unloaded at the same velocity until the force reached zero. The sample was then reloaded beyond the previous maximum load level, and this cycle was repeated until failure. The force and images of the bead pattern were recorded in the same manner as in the monotonic tensile test.

### Post-experiment analysis

The images of the fluorescent bead pattern were analyzed using commercial Digital Image Correlation software (Zeiss Inspect Correlate). Axial and transverse stretch data versus time were exported from the software. The width of the sample was measured from the confocal image of the undeformed sample using ImageJ software. The force per width, axial stretch, and transverse-axial stretch curves in all figures were smoothed using smoothing spline fitting with an R-squared value greater than 0.999. From these smoothed curves, the tangent stiffness and the incremental Poisson's ratio were calculated. The incremental Poisson's ratio is defined here as the negative ratio of increments of the natural logarithms of the stretches in the in-plane transverse and axial directions. The initial stiffness ($E_1$) and the initial Poisson's ratio ($v_1$) were determined as the average values of the tangent stiffness and incremental Poisson's ratio, respectively, in the stretch range from 1.02 to 1.05. The final stiffness ($E_2$) was calculated as the average value of the tangent stiffness from the stretch value where the derivative of the stiffness first approaches zero to the value as maximum stretch value minus 0.05. The peak Poisson's ratio ($v_2$) was defined as the average value of the incremental Poisson's ratio in the stretch range centered around the maximum incremental Poisson's ratio, specifically from the corresponding stretch value minus 0.05 to the value plus 0.05.

### Cross-polarization microscope imaging

We prepared the tensile test samples as described in the tensile test methods. For the stretched samples, we stretched them by hand under a dissecting microscope (Zeiss Stemi 508 with Excelis UHD camera), capturing images before and after stretching to calculate the stretch value. The two ends of the sample were then fixed on slides using tape to prevent any further deformation. The samples were imaged at both 0 degrees (stretching direction parallel to the polarizer) and 45 degrees (stretching direction at a 45-degree angle to the polarizer) under a cross-polarization microscope (Olympus BX60 microscope in transmission mode) using a Moticam 10+ camera paired with Motic Image Plus 3.0 software.

### Finite element simulations

We imaged the first true leaf from pLH13 (*p35S::mCitrine-RCI2A*) plants with a plasma membrane marker[96] using a Zeiss 710 confocal microscope and obtained the maximum intensity Z projection image. We constructed the cellular structures from the confocal images and stretched them in ABAQUS Standard (2018) as described in Supplementary Methods 3.1. For simplicity, we assumed the same thickness for all periclinal and anticlinal walls in the model. The thickness of the epidermal cell wall of Arabidopsis leaves ranges from approximately 0.6 μm to 1.5 μm[56], so we set the thickness of the cell walls to 1 μm in our cellular structure model. For comparison, thicknesses of 0.6 μm and 2 μm were also used. For the single plate model, the thickness was set equal to the combined thickness of the two periclinal walls in the cellular model, i.e., 2 μm, 1.2 μm, and 4 μm, respectively. Boundary conditions were prescribed as follows: $x = 0$ on boundary AB, $x = 0$ and $y = 0$ on boundary CD, and point D was fixed, and incremental displacement values were applied to the boundaries EF and GH (through defining equation constraints with a reference point R)

(Supplementary Fig. 10). Since we were applying a large deformation (maximum stretch was 1.5), we used the neo-Hookean model with Nlgeom (nonlinear geometry) turned on. Additionally, two Yeoh material models with different sets of parameters were used for comparison. We choose material parameters such that the initial modulus is equal to 1 and Poisson's ratio is 0.3. The initial modulus can be arbitrarily scaled as it only affects the magnitude of forces. The initial Poisson's ratio is chosen to fall within the experimental range. The material model parameters are shown in Supplementary Table S1. Finite element type is described in Supplemental Methods 3.2.

### Statistics

Statistical analyses were performed using MATLAB 2021a software. We considered $p$-values of less than 0.05 to be statistically significant. Individual data points are always shown with the mean ± standard deviation. A two-tailed unpaired Student's t-test was performed for the comparison between the two groups.

### Reporting summary

Further information on research design is available in the Nature Portfolio Reporting Summary linked to this article.

## Data availability

All data supporting the findings of this study are provided as a Source Data file, and are also available on Zenodo (https://doi.org/10.5281/zenodo.16460643). Source data are provided with this paper.

## Code availability

Custom MATLAB code was developed using MATLAB R2021a to implement the affine model and five-beam model, and the code is included in the Supplementary Software 1. ABAQUS Standard (2018) is used for finite element simulation of cellular structure, and the source file is included in the Supplementary Software 2. They are also available on Zenodo (https://doi.org/10.5281/zenodo.16460643). SENSIT 2.9 (Futek Inc.) was used for collecting force data. MATLAB R2021a was used to analyze the tensile test data.

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

## Acknowledgements

We thank Olivier Hamant for providing *spr2-2* and *qua2-1* seeds. We thank Alan Zehnder, Sarah Robinson, Virendra Puri, and M Shafayet Zamil for valuable discussions on the design of the micromechanical tensile stage. We thank Karl J. Niklas for insightful discussions on the mechanical behavior of plants. We thank Dorota Kwiatkowska for suggesting the use of fluorescent beads. We thank Catalin Picu for helpful discussions on mechanical behavior and models of fibrous network materials. We thank Mike Scanlon for sharing access to the high-temperature plant growth chamber. We thank Lanxi Hu and Ruqiang Zhang for their useful comments on the manuscript. Research reported in this publication was supported by the Engineered Living Materials Institute at Cornell University (S.C., B.P., M.N.S.), the National Institute of General Medical Sciences of the National Institutes of Health under Award Number R01GM134037 (I.B., A.H.K.R.), the National Science Foundation MCB-2203275 (A.H.K.R.), and the Sam and Nancy Fleming Postdoctoral Fellowship (S.C.). The authors acknowledge the use of facilities and instrumentation supported by NSF through the Cornell University Materials Research Science and Engineering Center DMR-1719875. The content is solely the responsibility of the authors and does not necessarily represent the official views of the National Institutes of Health.

## Author contributions

S.C.: conceptualization; methodology; formal analysis; investigation; writing - original draft; writing - review editing; visualization; funding acquisition. I.B.: methodology; resources; writing - review editing. P.J.: methodology; investigation; writing - review editing. B.P.: methodology; writing - review editing. M.N.S.: conceptualization; writing - original draft; writing - review editing; supervision; funding acquisition. A.H.K.R.: conceptualization; writing - original draft; writing - review editing; supervision; funding acquisition.

## Competing interests

The authors declare no competing interests.
