## [Transparent Peer Review file · Nature Communications]

Fibrous Network Nature of Plant Cell Walls Enables Tunable Mechanics for Development

Corresponding Author: Dr Adrienne Roeder

Version 0:

Reviewer comments:

Reviewer #1

(Remarks to the Author)

Thank you for giving me opportunity to read and review this manuscript which emphasises the role of fibrous assembly of the cell walls for plant tissue mechanics.

Novelty:

The main role for the cell wall and tissue mechanics has been assigned to cellulose-to-cellulose interactions that is in line with recent model of the cell wall assembly (the biomechanical hotspot model). However, more interpretation/discussion is needed on the nature of the cellulose-cellulose connections with consideration of existing literature: what is the mechanism that changes these connections (more comment below)? The experimental data are interpreted with relatively simple 5-beam model that considers bending of fibres and connectors mechanics, however recently at least two models (10.1126/science.abf28, 10.1016/j.carbpol.2024.121827) have been published based on the coarse grain molecular dynamic that are much more realistic to the real fibrillar structure and dimensions. These models provided similar but much more deep conclusions on the role of cellulose fibres in cell wall, i.e., it was showed that the mechanics of a composite made of fibres relates to fibre interaction, fibre modulus, fibre length (number of inter-fibre links). Thus, the above conclusion obtained in proposed manuscript I cannot consider as novel.

Neglecting by authors the cellular geometry (micro-scale level) on tissue mechanics, and actually generalization of this observation, has not been sufficiently discussed in this manuscript. Taking into account that this conclusion bases on FEM models, without such extended discussion I cannot consider as sufficiently proved (see comment below).

Organisation of the manuscript:

I appreciate very clear presentation of results in the main text and moving details to supplementary. However, I think that the link between the main article and supplementary of results would be improved by giving in the main text references to specific fragments/paragraphs in the supplementary. It will improve readability and understanding the approach and design of the experiment.

Major drawbacks:

(1) A five-beam model presented in this manuscript, although very simple, provides acceptable (logical) conclusion that the interaction of the cellulose fibres (mimicking the biomechanical hotspots) is crucial for mechanical properties of the cell wall and tissue. However, in this work there is lack of interpretation of the connector between cellulose, although assumed mechanics of connector one may find in supplementary. Please speculate or provide evidences about the physico-chemistry of the cellulose-cellulose contacts?. What would be the mechanism of modification of the cellulose-cellulose contacts (shear stress) if the matrix polysaccharides have just little effect, as stated in conclusions?. Please note that there are many evidences published that cellulose interacts with hemicelluloses and pectin indirectly or directly, so please refer to them (examples: 10.1104/pp.105.065912, 10.1021/bi3015532, 10.1080/10408398.2013.850652, 10.1016/j.pbi.2018.07.016).

(2) It is concluded, based on comparison of two FEM models (flat surface vs generated tissue structure with features of epidermal membrane of Arabidopsis), that cellular structure does not determine the shape of the force-stretching curve. Indeed, video S3 shows that during stretching only the top and bottom walls are stretched while cell walls oriented

perpendicularly to the force deform just slightly. However, this is very specific case. For example, for turgid, convex and spherical parenchyma cells with low number of spaces (highly compact structure), deformation may be immediately transferred to cell wall stretching (10.1016/j.jfoodeng.2013.09.012). In such a case, if more and more walls are reoriented and stretched it would cause nonlinear stiffening of the tissue too. So, in such a case, the force curve is very dependent on the cellular structure. Therefore, conclusion that “geometry of cellular structure is only minor factor in the force deformation response” may be specific to the system studied and cannot be generalized, taking into account variability of cell geometries existing in nature.

(3) Using N/m instead of the stress units (Pa) for experimental data may lead to incorrect interpretations if thickness of epidermis membrane varies between samples. Please kindly provide additional data: (a) scattering of the membrane thickness for biological replicates, (b) the membrane thickness for different development stages and for the mutant used. Consequently, please change the graphs by replacing N/m by Pascal unit.

(Remarks on code availability)

Reviewer #2

(Remarks to the Author)

In this paper, the authors assess the biomechanical properties of plant cell walls during growth. Per my expertise, I have limited my review to the biomechanics, and will defer to my fellow reviewers for other issues (e.g. physiology, genetics, etc.). This work is important, and has the potential to be very impactful. I would like to specifically commend the authors on their interdisciplinary approach, bringing in authors from both the engineering and biological sciences to present a well-thought out research paper. In our field of computational plant biomechanics, true interdisciplinary collaboration is paramount to push this field forward, and the authors have done a great job. Overall, this paper is excellent.

My only minor comments are with respect to the finite element model:

1. In the SI, you mention that the shell thickness corresponds to the thickness of the measured walls, but in the model, the thickness is simply set to 1. Was this intentional (i.e. was the wall thickness actually found to be 1 μ m)?
2. In the SI, Figure 7 depicts the boundary conditions. I would also suggest visually depicting or adding in to the caption that an equation constraint was used to link the top and bottom surfaces to the reference point being moved.
3. Although not needed for this study, for follow-on / future investigations into the FEM may I suggest:
 - 3a. The model could be simplified by using a symmetry boundary condition at the halfway plane between the top and bottom surfaces, which would reduce the computational time and simplify the required boundary conditions.
 - 3b. Merging the three part instances would ensure a better-aligned mesh and remove the need for a tie constraint, which can introduce increases in local stiffness and stress values.

(Remarks on code availability)

I reviewed the *.cae file in its entirety. No further documentation is required.

Reviewer #3

(Remarks to the Author)

This manuscript by Chen et al. examines the mechanical behaviour of Arabidopsis leaf epidermal cell walls. The authors use micromechanical tensile tests, confocal microscopy, and theoretical modelling to describe a three-phase non-linear stress–strain response, a non-linear Poisson effect, and progressive plastic deformation. They also analyse how these properties change during development and compare wild-type plants with the spr2-2 mutant. The experimental design is solid. The combination of tensile tests on epidermal peels with confocal microscopy allows for precise measurements of forces, stretches, and deformations. The data clearly show three mechanical regimes: an initial low-stiffness phase, a strain-stiffening phase, and a high-stiffness plateau. Tracking Poisson’s ratio across a broad range provides useful insights, showing that it varies with stretch rather than remaining constant.

The theoretical framework is well supported by finite element simulations and a five-beam model, which explain how cellulose microfibrils transition from bending-dominated to stretch-dominated behaviour. However, the link between model parameters and experimental data could perhaps be made clearer. A more detailed discussion of the model’s assumptions and limitations would also be helpful.

The use of the spr2-2 mutant is interesting, as its altered microtubule dynamics influence cellulose microfibril organisation and thus cell wall anisotropy. The anisotropy analysis supports the idea that changes in microfibril organisation affect stiffness. However, since spr2-2 affects cell wall mechanics indirectly, it would be useful to include mutants that more directly impact cell wall synthesis, such as cellulose synthase mutants, or those that alter pectin modification. This would provide stronger evidence for the proposed link between microfibril connectivity and stiffness changes.

Tracking mechanical properties in the context of plant development is a very interesting approach that is not that common in this type of papers, adding an important insight. However, a clearer description of the specific changes occurring in plant cell walls over time would strengthen the study. From the biological point of view, a more detailed account of how cell wall

composition, microfibril connectivity, and organisation evolve and how these changes influence mechanical properties would make the developmental analysis more informative.

Overall, the manuscript provides valuable insights into plant cell wall mechanics, but major revisions are needed. My recommendations to the authors are:

1. Strengthen the quantitative link between the theoretical model and the experimental data.
2. Expand the mutant analysis to include systems that more directly affect cell wall synthesis or pectin modification.
3. Provide a clearer discussion of specific developmental changes in cell wall structure and composition.

Finally, I would like to congratulate the authors. I believe this is a strong contribution to the field, and with some revisions, it has the potential to make a great impact.

(Remarks on code availability)

Reviewer #4

(Remarks to the Author)

In the present manuscript, the authors investigate the mechanical behavior of the leaf epidermis of the first true leaves of *Arabidopsis* through tensile testing. They use a series of complementary approaches to derive information on the behavior of the fibrous network material composing the cell wall. They focus in particular on the non-linearity of the stretching behavior and on Poisson's ratio, a property that describes the behavior of the material in a direction transverse to the load application. Through *in silico* modeling, the experimental results are interpreted and conclusions are drawn as to the role during plant developmental processes.

Overall, this is a very carefully executed study that is meticulously documented and provides important insight into the mechanical behavior of plant cell wall material in a growing organ. The insight is novel and the conclusions provide important new information.

I have a few concerns, however, which the authors may wish to consider.

FUNDAMENTAL

1. Line 165: Determining Poisson's ratio of a material presumably requires for the flat strip of tissue to remain flat during axial load application. However, thin membranous material like the one tested here has the tendency to form wrinkles parallel to the load application to relax the compressive force resulting from the Poisson effect. Wrinkle formation presumably complicates the quantification of Poisson's ratio of the material property (vs the overall structure). Poisson's ratio of the epidermis material should be independent of the experimental setup of the tensile test, whereas the measured value for Poisson's ratio of a material wrinkling under tensile strain is presumably influenced by the exact setup and sample dimensions. It would therefore be crucial to validate whether wrinkles arise upon stretching of the samples examined here and in the specific setup and sample geometry used. This could be done by performing 3D confocal imaging on the stretched material and doing 3D reconstruction. Since wrinkle formation would potentially change the entire interpretation of a key message in this paper, I consider attention to this aspect a fundamental point that would need to be addressed.

Related, Figure 1H suggests that the behavior of the material is visco-elastic. There seems to be little attention to this observation in the interpretation of the data. It could have significance for Poisson's ratio, since the ratio may show time-dependent behavior as well - and not necessarily in the same manner as the recovery from strain in axial direction. This highlights the importance of standard procedures (in terms of temporal behavior) when measuring Poisson's ratio in visco-elastic material. The authors may wish to consider this.

2. Interesting results regarding the anisotropy of the tensile behavior of the epidermis were made in the mutant *spr2-2*. In the discussion, the mechanical anisotropy is related to the 'enhanced alignment of microfibrils' (line 562) which is reasonable but no proof is provided. I suggest that experimental evidence for the same be added to the manuscript. Standard staining protocols for cellulose are readily available (e.g. Bidhendi et al 2020. *Journal of Microscopy* 278: 164-181) and would significantly strengthen the Conclusions.

MAJOR

3. While much of the relevant literature is cited, the authors may wish to consider referring to Rosemary Dyson's work, e.g. doi.org/10.1016/j.jtbi.2021.110736 (in the context of page 8) or this book chapter (<https://escholarship.mcgill.ca/concern/articles/q811kr12s>)

4. Line 140: From this description it appears that the epidermal peel consists of the intact epidermal cell layer meaning that it comprises the upper and lower periclinal walls and the anticlinal walls, and that the epidermal cells are intact (albeit deflated). If this is correct, this fact should be spelled out very clearly since in the much used onion epidermal peels used by others, the peel consists only of half of the epidermal layer since during stripping, the anticlinal walls rupture (whereas in the present manuscript the rupture seems to be at the connection between mesophyll and epidermis). In the same context: Was it consistently the abaxial epidermis that separated when performing the manoeuvre shown in Fig 5? This might be worth mentioning.

5. Also, here and elsewhere it would help if the term 'periclinal wall' was used throughout where appropriate.
6. Line 146: I find the term 'membrane force' an unfortunate choice in the present context. I understand that it is probably taken from the engineering language, but for a biologist, the term 'membrane' has such a dominant, very different significance that it keeps distracting me. While it is up to the authors, I suggest finding a different term (sheet force?).
7. Line 146: What the authors define to be the 'axial stretch' is really an axial 'strain' in engineering terms. I suggest using that terminology instead.
8. Line 150: Is the test strip truly under tension in Regime I or is it relaxing tissue folds (oriented transverse to the load application) that might have been created during the excision process?
9. Line 245: What are the dimensions or properties of the 'single plate'? A plate with thickness (or stiffness) twice that of a single periclinal wall plate in the cellular model? Please specify.
10. Line 249 and Fig 2B: Neither from the figure nor the text is it truly clear how the FE model is structured. Do the anticlinal walls connect to the outer and inner periclinal sheets? Is the lumen of the cells empty? Does the lumen have properties? Also, was the fact that in the biological system the anticlinal walls are thinner than the periclinal walls of the epidermis, and that the out periclinal wall is thicker than the inner periclinal wall, considered in the model? I don't think it would make much of a difference in the outcome here, but it would truly be helpful to specify the simplifying assumptions much more clearly, certainly where they deviate from biological reality.
11. Line 262: I suggest adding '...from the material PROPERTIES of the PERICLINAL cell walls'. If this is not what is meant here, please explain the statement better in different manner.
12. Line 274: While the microfibrils in the periclinal walls of the Arabidopsis leaf epidermis are indeed isotropically oriented IN AVERAGE or OVERALL, in these wavy pavement cells there is significant complexity at subcellular level with instances of anisotropic orientation (Altartouri et al 2019 Plant Physiology 181: 127–141). This fact should at least be mentioned.
13. Line 276: This is a very well known effect, and in the context of plant cell walls has been proposed as early as 1953 (multinet growth theory by Roelofsen and Houwink (1953) Acta Botanica Néerlandica 2:218–125). Mention of this would be appropriate in order to give credit to early thinkers.
14. Lines 416-417: Although it is not intended this way, this sentence may suggest to non-expert readers that plant cell growth is driven by an increase of turgor. That notion would be incorrect, of course. The trigger for growth is a reduction in cell wall stiffness instead, leading to a drop in turgor that in turn leads to an uptake of water that stretches the wall. Maybe the statement could be reworded somewhat.

MINOR

15. Lines 81 & 315: 'It's' is inappropriately informal and should be 'it is'
16. Line 139: Although it is mentioned in the Methods section, I suggest mentioning here the exclusively the first true leaves are used for testing. Also, the orientation and location of the standard test strip should be described or, even better, shown as a drawing or photograph.
17. Figure legend 1C: Specify whether these are single optical sections or projections of a 3D stack.
18. Lines 253 and 256 I suggest adding '...deviation WITHIN THE TESTED RANGE OF DEFORMATION'
19. Line 458: I suggest specifying the plant organ that was investigated in the study by Hervieux et al.
20. Line 574: I suggest adding '...different organs, species AND GROWTH STAGES'

(Remarks on code availability)

Version 1:

Reviewer comments:

Reviewer #1

(Remarks to the Author)

Thank authors for addressing my comments and corrections in the manuscript.

Still one point must be improved:

The explanation regarding my comment on “using N/m instead of the stress units (Pa)” is speculative and hard to understand. Following the explanation, I agree that units in the equations are ok. But, the explanation refers to relation of turgor (in single cell) to force membrane (of the tissue) while cells are not turgid in this case (specific experiment to confirm this fact has been added). To clarify your way of thinking, I suggest to present detailed explanation related to this problem in Supplementary Materials. On the other hand, it is very intuitive that in the experimental stretching the sample cross-section (not only width) determines the force detected on the sensor. I must admit that I do not have experience in studying epidermis and my suspicious about varying thickness related to development stage and mutant is maybe not very important. But, if the thickness of the stretched stripe changes substantially, it would not mean that the material property (stiffness) of the epidermis changes at all while the measured force-strain change results from change of thickness. Therefore, I consider my question and authors explanation regarding this point as fundamental for this work (hopefully explained in details in supplementary and more clearly than in the response to my request).

(Remarks on code availability)

Reviewer #2

(Remarks to the Author)

All comments have been adjudicated. I sincerely thank the authors for their thorough and thoughtful responses.

(Remarks on code availability)

Reviewer #3

(Remarks to the Author)

This revised manuscript addresses the key points raised in my previous review and improves the clarity and biological relevance of the findings. Overall, the authors have responded thoroughly. The revised manuscript presents a more complete account of how wall composition and architecture relate to nonlinear mechanics in epidermal tissues. The modelling, mutant analysis, and developmental comparisons are now more clearly connected.

In response to my and other reviewers' comments, the authors have strengthened the connection between their theoretical models and experimental data. The finite element simulations are now more clearly described, including how cell geometries were extracted and how turgor loss and wall thickness were handled. The assumptions about wall structure and material properties are now explicitly stated. While the model still simplifies subcellular features, the authors have acknowledged its limitations and clarified its intended use.

The five-beam model is also better justified. Parameter choices are supported by literature-based estimates of cellulose microfibril geometry and spacing, and the model's capacity to capture distinct deformation modes is now more clearly explained. The authors have added a discussion of the model's simplifying assumptions, as requested.

My earlier suggestion to include additional mutants affecting wall synthesis or matrix composition has been addressed. The addition of qua2-1, which reduces homogalacturonan content, is appropriate and informative. The mechanical results from qua2-1 show that early stiffness is unaffected, while rupture occurs earlier than in wild type, consistent with compromised cell adhesion or wall cohesion. This strengthens the conclusion that pectin contributes more to strength at high strain than to initial stiffness. Although the rsw1-1 data are not included in the manuscript, I appreciate the authors sharing the results in the rebuttal. The mechanical similarity to wild type at the tested temperature is interesting and may warrant further study, though it is reasonable to leave this outside the scope of the current manuscript.

The discussion of developmental changes in wall composition has been expanded to include relevant biological processes, such as pectin de-esterification and matrix polysaccharide integration. These additions make the developmental aspect of the study more informative. However, the term “microfibril connectivity,” used to explain the increase in stiffness over time, remains broad. It would be helpful if the authors could clarify whether this refers to tighter bundling, greater inter-fibril adhesion, matrix cross-linking, or another structural feature. Even a conceptual definition would help ground the interpretation.

It might also be useful to consider whether the observed changes in plasticity or recovery could provide indirect support for reduced slippage, if no direct structural evidence is available. Additionally, it would be helpful to note whether the findings generalise beyond the sampled leaf ages and positions, or if they are limited to the specific developmental stages examined.

In addition to addressing my own comments, the authors have incorporated relevant revisions in response to points raised by the other reviewers. These include clearer treatment of anisotropy and imaging limitations, justification of force units, clarification of modelling terminology, and a more explicit discussion of assumptions. These revisions further strengthen the manuscript and improve its clarity.

(Remarks on code availability)

Reviewer #4

(Remarks to the Author)

The authors have addressed or rebutted most of my previous comments satisfactorily. It is a fantastic work, congratulations.

A few minor concerns remain:

Lines 270, 424 and potentially elsewhere: I think that typically, there would be no article ('the') before Poisson's ratio, similar to, say, 'Hooke's law'.

Line 756: 'It's' is inappropriately informal and should be 'it is'.

The authors omitted addressing or rebutting two of my previous comments:

19. Line 458 (now line 532): I suggest specifying the plant organ that was investigated in the study by Hervieux et al.

20. Line 574 (now line 645): I suggest adding '...different organs, species AND GROWTH STAGES'

(Remarks on code availability)

Reviewer #1 (Remarks to the Author):

Thank you for giving me opportunity to read and review this manuscript which emphasises the role of fibrous assembly of the cell walls for plant tissue mechanics.

We thank the reviewer for a careful evaluation of the manuscript, and we appreciate the valuable comments the reviewer provides. We addressed the comments in the following response and accordingly in the revised version.

Novelty:

The main role for the cell wall and tissue mechanics has been assigned to cellulose-to-cellulose interactions that is in line with recent model of the cell wall assembly (the biomechanical hotspot model). However, more interpretation/discussion is needed on the nature of the cellulose-cellulose connections with consideration of existing literature: what is the mechanism that changes these connections (more comment below)?

We appreciate the reviewer's thoughtful and detailed comments regarding cell wall modeling approaches and the mechanisms of connections between cellulose microfibrils. We have expanded our discussion of cellulose-cellulose interactions by incorporating relevant literature in the revision. Detailed responses are provided to the comments below.

The experimental data are interpreted with relatively simple 5-beam model that considers bending of fibres and connectors mechanics, however recently at least two models (10.1126/science.abf28, 10.1016/j.carbpol.2024.121827) have been published based on the coarse grain molecular dynamic that are much more realistic to the real fibrillar structure and dimensions. These models provided similar but much more deep conclusions on the role of cellulose fibres in cell wall, i.e., it was showed that the mechanics of a composite made of fibres relates to fibre interaction, fibre modulus, fibre length (number of inter-fibre links). Thus, the above conclusion obtained in proposed manuscript I cannot consider as novel.

We agree that coarse grain molecular dynamic simulations have provided significant insights into cell wall mechanics with consideration of detailed and realistic structure, and we have added these studies into our discussion. Our modeling approach offers a complementary perspective to these MD studies: a simplified and computationally efficient model that still captures the key features observed experimentally. The five-beam model intuitively demonstrates the mechanisms controlling mechanical behavior, which is the progression of cellulose microfibrils from reorientation and bending-dominated to stretch-dominated deformation modes. Furthermore, the model not only explains a single case of deformation, but

also provides insights into how mechanical behavior changes during development and in microtubule mutants. We believe that our model is valuable for advancing understanding of plant cell wall mechanics during development.

In addition, we would like to emphasize that our manuscript provides novel experimental findings. To our knowledge, the three-regime nonlinear mechanical behavior observed in our experiments has not been previously reported for plant cell walls. This behavior notably differs from the mechanical behaviors commonly reported in onion epidermal cell walls. Our findings, therefore, provide new insights into the diversity of mechanical properties among plant tissues that share similar components. Moreover, our study links cell wall mechanics to plant growth and development, by exploring changes in mechanical behavior among different development stages and in a microtubule dynamics mutant, which contributes to broader biological insights.

Neglecting by authors the cellular geometry (micro-scale level) on tissue mechanics, and actually generalization of this observation, has not been sufficiently discussed in this manuscript. Taking into account that this conclusion bases on FEM models, without such extended discussion I cannot consider as sufficiently proved (see comment below).

We appreciate the reviewer's valuable comments on cellular geometry on tissue mechanics. We have included an expanded discussion on this aspect. We have clarified under what conditions our conclusions remain valid and included references showing cases where cellular geometry does significantly affect mechanical behavior. Detailed responses are provided to the comments below.

Organisation of the manuscript:

I appreciate very clear presentation of results in the main text and moving details to supplementary. However, I think that the link between the main article and supplementary of results would be improved by giving in the main text references to specific fragments/paragraphs in the supplementary. It will improve readability and understanding the approach and design of the experiment.

We appreciate the reviewer's comment on the presentation style. To improve readability and strengthen the connection between the main text and supplementary materials, we have added explicit references to specific paragraphs in the supplementary sections.

Major drawbacks:

(1) A five-beam model presented in this manuscript, although very simple, provides acceptable (logical) conclusion that the interaction of the cellulose fibres (mimicking the biomechanical hotspots) is crucial for mechanical properties of the cell wall and tissue. However, in this work there is lack of interpretation of the connector between cellulose, although assumed mechanics of connector one may find in supplementary. Please speculate or provide evidences about the physico-chemistry of the cellulose-cellulose contacts?. What would be the mechanism of modification of the cellulose-cellulose contacts (shear stress) if the matrix polysaccharides have just little effect, as stated in conclusions?. Please note that there are many evidences published that cellulose interacts with hemicelluloses and pectin indirectly or directly, so please refer to them (examples: 10.1104/pp.105.065912, 10.1021/bi3015532, 10.1080/10408398.2013.850652, 10.1016/j.pbi.2018.07.016).

We think both direct cellulose-cellulose contact and matrix polysaccharides could be the physico-chemical nature of the connector element between cellulose microfibrils, and changes in the connector could come from modifications to either or both. To address the reviewer's concern, we have expanded the discussion in **Section 2.2** and **3** to include potential physico-chemical mechanisms underlying these connections, referencing cellulose-cellulose contact as well as the role of hemicellulose and pectin. We have also incorporated the studies suggested by the reviewer, along with additional literature, to provide a more comprehensive view. In addition, we have conducted experiments using *qua2-1*, a mutant with around 50% reduction in homogalacturonan (HG) content compared with the wild type (Mouille et al, 2007, doi: 10.1111/j.1365-313X.2007.03086.x). We found that the initial stiffness (stretch from 1 to 1.05) and the early strain-stiffening behavior (stretch from 1.05 to 1.1) of epidermal peels were comparable to the wild type while the stiffening regime was terminated earlier by softening and rupture (**Figure S7**). We think this result suggests that HG reduction does not significantly affect cellulose microfibril connections during early deformation but may compromise connection strength and cell adhesion. We have included these results and their interpretation in **Section 3**.

The revised text and added **Figure S7** are included below for reference:

In **Section 2.2**, "...These links transfer mechanical forces from one microfibril to another. Their physical nature could result from direct cellulose-cellulose contact, interactions mediated by the polysaccharide matrix, or both [10]..."

In **Section 3**, "...The physical nature of the connectors may arise from the polysaccharide matrix and direct cellulose-cellulose contacts. Xyloglucan, the dominant hemicellulose in the cell wall, can bind to cellulose microfibrils by hydrogen bonds or be trapped in the microfibrils [83]. Although evidence suggests that the majority of xyloglucan does not contribute to wall mechanics, a small fraction might intertwine with cellulose microfibrils to form junctions controlling wall mechanics, known as the "biochemical hotspots" hypothesis [22]. In addition, pectin with neutral sugar side chains can bind cellulose, likely through hydrogen bonds, potentially providing linkages [84, 85]. However, it remains unclear how strong this interaction is [86]. Direct cellulose-cellulose contacts have recently received more attention, suggesting they

might be the strongest connections transferring tensile forces within the cellulose network [32, 87]...”

“There are several possible physical mechanisms causing alterations in the connections between cellulose microfibrils during plant development. One possibility is modification of the polysaccharide matrix. For example, changing the concentration of hemicellulose influences the tensile mechanical properties of a cell-wall analog material [88, 89]. During growth, newly secreted matrix polysaccharides, such as pectin and hemicellulose, are integrated into the existing cellulose network and may enhance connectivity between cellulose microfibrils [91]. As cells mature, homogalacturonan (HG), a major pectin component, becomes less methyl-esterified, allowing more calcium ionic crosslinking and resulting in stiffer pectin gels [92]. To assess whether reduced HG content affects mechanical behavior, we performed tensile tests on *qua2-1* (SI Methods E, Other plant material), a mutant with around 50% less HG and defective cell adhesion [90, 91]. We found that the initial stiffness (stretch from 1 to 1.05) and the early strain-stiffening behavior (stretch from 1.05 to 1.1) were comparable to the wild type. However, the stiffening regime ended earlier due to softening and rupture (Figure S7). This suggests HG reduction does not significantly affect cellulose microfibril connections during early deformation but may compromise the strength of these connections and cell adhesion. Additionally, direct cellulose–cellulose interactions may play a more critical role in altering connections between cellulose microfibrils compared to matrix polysaccharides. A coarse-grained physical model of epidermal cell walls [32] shows that strengthening cellulose–cellulose contacts increases wall stiffness and decreases plasticity while increasing the binding energies between cellulose and matrix polysaccharides has little effect. These cellulose–cellulose contacts can be influenced by factors such as water and xyloglucan[92]...”

Figure S7

Figure S7: Comparison of monotonic tensile test results for wild type and *qua2-1*. Force per width – axial stretch and transverse – axial stretch curve for wild type at 25 days

post-germination and *qua2-1* at 26 days post-germination. Note that five curves from wild type are reused from Figure 3B 25d data.

(2) It is concluded, based on comparison of two FEM models (flat surface vs generated tissue structure with features of epidermal membrane of Arabidopsis), that cellular structure does not determine the shape of the force-stretching curve. Indeed, video S3 shows that during stretching only the top and bottom walls are stretched while cell walls oriented perpendicularly to the force deform just slightly. However, this is a very specific case. For example, for turgid, convex and spherical parenchyma cells with low number of spaces (highly compact structure), deformation may be immediately transferred to cell wall stretching (10.1016/j.jfoodeng.2013.09.012). In such a case, if more and more walls are reoriented and stretched it would cause nonlinear stiffening of the tissue too. So, in such a case, the force curve is very dependent on the cellular structure. Therefore, conclusion that “geometry of cellular structure is only minor factor in the force deformation response” may be specific to the system studied and cannot be generalized, taking into account variability of cell geometries existing in nature.

We appreciate the reviewer’s insightful comment regarding the influence of cellular structure on force-deformation behavior. We agree that cell turgidity influences the mechanical behavior of the tissue, and we conducted additional experiments in which we froze and thawed the tissue to verify that the cells are not turgid in our case. This has been added to **Section 2.2**. We agree that the extent to which cellular geometry affects mechanical response depends on specific cell geometries. We have expanded the discussion in **Section 3** to clarify the conditions under which the conclusion that “the geometry of cellular structure is only a minor factor in the force-deformation response” is valid. Additionally, we have referenced relevant literature showing cases where cellular structure impacts force-deformation behavior.

The revised text is included below for reference:

Section 2.2, “...After peeling, the epidermal cells lose their turgor pressure (Figure S1B). We further verified this by performing tensile tests on the frozen and thawed treatment sample (SI Method - Leaf epidermal peel preparation), and we showed mechanical behavior similar to that of untreated samples (Figure S1C). Therefore, in our cellular structure model, the top and bottom periclinal cell walls were treated as flat plates...”

Section 3, “...In our study of peeled Arabidopsis leaf epidermis, the cellular structure has limited influence on the tensile mechanical behavior at the tissue level, as indicated by our finite element simulation. This is because pavement cells are deflated and have very wavy anticlinal walls, which offer minimal mechanical resistance during stretching. In many eudicot plants, pavement cells have this jigsaw puzzle shape [8], suggesting that this finding could also apply to these cases. However, this conclusion may not extend to tissues composed of polygonal cells, such as the onion epidermis, where the anticlinal wall could be reoriented and stretched when

the tissue is deformed [67, 68]. In such cases, the cellular structure is more likely to influence overall tissue mechanics...

(3) Using N/m instead of the stress units (Pa) for experimental data may lead to incorrect interpretations if thickness of epidermis membrane varies between samples. Please kindly provide additional data: (a) scattering of the membrane thickness for biological replicates, (b) the membrane thickness for different development stages and for the mutant used. Consequently, please change the graphs by replacing N/m by Pascal unit.

We appreciate the reviewer's comment regarding the choice of N/m over stress units (Pa), this is something we have discussed amongst ourselves extensively. Our primary reason for selecting membrane force (N/m, now termed "force per width" in the main text following a suggestion from Reviewer 4) rather than stress (Pa) is that it allows a direct connection between experimental measurements and the biophysical model that relates turgor pressure to cell wall deformation. The membrane force-stretch relationship from experimental data is: membrane force = stiffness * (stretch - 1), where stiffness is a function of stretch. Within a cell, the balance of forces can be expressed as: turgor pressure * cross section area = membrane force * perimeter of cross section. Therefore, we have turgor pressure relationship to stretch is: turgor pressure = membrane force * perimeter of cross section / cross section area = stiffness * (stretch - 1) * perimeter of cross section / cross section area. This formulation incorporates experimental results without requiring explicit thickness measurements. In addition, we note that estimating cell wall thickness for each sample introduces uncertainty. The thickness varies across different locations within the cell, for example, the wall is typically thicker near cell boundaries than at the center, making it difficult to determine a consistent average thickness. Finally, our conceptual findings are not affected by our choice to use membrane force rather than stress. When comparing mechanical properties across different developmental stages, prior work has shown that cell wall thickness does not change much during cell growth (Lockhart 1965, 10.1016/0022-5193(65)90077-9), thus values in N/m reflect a meaningful trend of mechanical properties. For comparing mechanical properties in different directions, both in wild type and *spr2-2* mutant, our primary focus is on anisotropy, rather than comparing absolute values between *spr2-2* and wild type.

At the same time, we recognize that reporting stress-based modulus values (MPa) may provide additional insights for readers. To address this, we have calculated the modulus based on assuming effective cell wall thickness and included this information in **Section 2.1**.

The revised text is included below for reference:

"..In regime I, the sample is relatively soft, with a nearly constant low level of stiffness (~ 12

N/m, which is equivalent to 12 MPa if an effective wall thickness of 1 μ m is assumed, based on the outer periclinal wall being about 1 μ m thick [56] and the inner periclinal wall being too thin to be considered [57]). Then in regime II, the stiffness gradually increases as the stretch increases,

exhibiting a nonlinear strain-stiffening behavior. This mechanical behavior suggests that more “elements” within the sample are becoming active in resistance to the loading. Finally, in regime III, the tissue is relatively stiff, with a nearly constant tangent stiffness (~ 70 N/m, which is equivalent to 70 MPa) (Figure 1D-E). The stiffness range is similar to that reported in tensile tests on Arabidopsis hypocotyls [40].”

Reviewer #2 (Remarks to the Author):

In this paper, the authors assess the biomechanical properties of plant cell walls during growth. Per my expertise, I have limited my review to the biomechanics, and will defer to my fellow reviewers for other issues (e.g. physiology, genetics, etc.). This work is important, and has the potential to be very impactful. I would like to specifically commend the authors on their interdisciplinary approach, bringing in authors from both the engineering and biological sciences to present a well-thought out research paper. In our field of computational plant biomechanics, true interdisciplinary collaboration is paramount to push this field forward, and the authors have done a great job. Overall, this paper is excellent.

We thank the reviewer for the comments. We appreciate the recognition of the interdisciplinary nature of our work. We have addressed the comments in the following response and accordingly in the revised version.

My only minor comments are with respect to the finite element model:

1. In the SI, you mention that the shell thickness corresponds to the thickness of the measured walls, but in the model, the thickness is simply set to 1. Was this intentional (i.e. was the wall thickness actually found to be 1 μ m)?

We thank the reviewer for this important point. Based on the literature, the thickness of the cell wall typically ranges from 0.6 to 1.5 μ m. In the model, we used 1 μ m as a representative value. We have stated it explicitly in the **Method - Finite Element Simulation** of the main text in the revision. In addition, we have included simulation results with varying thickness values in **Figure S4D-E**. For model simplicity, we assumed the thickness is uniform everywhere, and we have added a discussion of this assumption in the **Section 3**.

The revised text is included below for reference:

Method - Finite Element Simulation: *“The thickness of the epidermal cell wall of Arabidopsis leaves ranges from approximately 0.6 μ m to 1.5 μ m [87], so we set the thickness of the cell walls to 1 μ m in our model. For comparison, thicknesses of 0.6 μ m and 2 μ m were also used.”*

Section 3: *“...In addition, we note that subcellular details are simplified in our FEM model, where we assume uniform thickness and homogeneous mechanical properties for both periclinal and anticlinal walls. Although we believe these simplifications do not substantially impact our main conclusions, they should be reconsidered if precise deformation patterns at the subcellular scale are of particular interest...”*

2. In the SI, Figure 7 depicts the boundary conditions. I would also suggest visually depicting or adding in to the caption that an equation constraint was used to link the top and bottom surfaces to the reference point being moved.

We have added this description to the caption of **Figure S10** (formerly Figure S7) and also modified the labels in **Figure S10** to explicitly state that an equation constraint was used and the reference point is being moved.

3. Although not needed for this study, for follow-on / future investigations into the FEM may I suggest:

3a. The model could be simplified by using a symmetry boundary condition at the halfway plane between the top and bottom surfaces, which would reduce the computational time and simplify the required boundary conditions.

3b. Merging the three part instances would ensure a better-aligned mesh and remove the need for a tie constraint, which can introduce increases in local stiffness and stress values.

We thank the reviewer for these useful suggestions. Using symmetry boundary conditions at the halfway plane and part merging are great approaches, and we will incorporate them in our future modeling tasks.

Reviewer #2 (Remarks on code availability):

I reviewed the *.cae file in its entirety. No further documentation is required.

We thank the reviewer for reviewing the model files.

Reviewer #3 (Remarks to the Author):

This manuscript by Chen et al. examines the mechanical behaviour of Arabidopsis leaf epidermal cell walls. The authors use micromechanical tensile tests, confocal microscopy, and theoretical modelling to describe a three-phase non-linear stress–strain response, a non-linear Poisson effect, and progressive plastic deformation. They also analyse how these properties change during development and compare wild-type plants with the spr2-2 mutant. The experimental design is solid. The combination of tensile tests on epidermal peels with confocal microscopy allows for precise measurements of forces, stretches, and deformations. The data clearly show three mechanical regimes: an initial low-stiffness phase, a strain-stiffening phase, and a high-stiffness plateau. Tracking Poisson's ratio across a broad range provides useful insights, showing that it varies with stretch rather than remaining constant.

We thank the reviewer for the detailed and thoughtful feedback. We have addressed the comments in the following response and accordingly in the revised version.

The theoretical framework is well supported by finite element simulations and a five-beam model, which explain how cellulose microfibrils transition from bending-dominated to stretch-dominated behaviour. However, the link between model parameters and experimental data could perhaps be made clearer. A more detailed discussion of the model's assumptions and limitations would also be helpful.

We appreciate this useful comment. First, in the finite element simulations, we reconstructed the realistic cellular structure based on actual cell geometries and arrangements in the leaf epidermis. Specifically, the anticlinal wall outlines were extracted from confocal images. The periclinal walls were modeled as flat plates, reflecting the loss of turgor pressure after peeling. The loss of turgor pressure is supported by the frozen and thawed treatment experiments, now included in **Section 2.2**. The anticlinal and periclinal walls were connected via tie constraints to form the final cellular structure. The cell lumen was modeled as empty, again reflecting the absence of turgor pressure. The thickness of both anticlinal and periclinal walls was set to 1 micron, which falls within the reported range (0.6 – 1.5 microns, 10.1111/tpj.12042). We also provided simulation results with varying thicknesses in **Figure S4D-E**. For the wall material, we used a neo-Hookean model, a commonly used and relatively simple model for large deformation simulations. To assess sensitivity to material behavior, we also tested two Yeoh material models with strain-softening and strain-stiffening behavior respectively, and results are included in **Figure S4F-G**. For neo-Hookean and Yeoh models, we choose parameters such that the initial modulus is equal to 1 and Poisson's ratio is 0.3. The initial modulus can be arbitrarily scaled as it only affects the magnitude of forces. The initial Poisson's ratio is chosen to fall within the experimental range. We have added a more detailed explanation of the FEM model in **SI Methods - Extracting the cellular structure from the confocal images** and **Section 4 Methods - Finite element simulations** in our revised paper. While we assume uniform cell wall thickness and homogeneous mechanical properties across the tissue, we believe these

assumptions are sufficient to prove that cellular structure has limited influence on overall mechanical behavior. A discussion of modeling assumptions has also been added to **Section 3**.

The revised text is included below for reference:

Section 2.2, “...After peeling, the epidermal cells lose their turgor pressure (Figure S1). We further verified this by performing tensile tests on the frozen and thawed treatment sample (SI Method - Leaf epidermal peel preparation), and we showed mechanical behavior similar to that of untreated samples (Figure S1). Therefore, in the cellular structure model, the top and bottom periclinal cell walls were treated as flat plates...”

SI Methods - Extracting the cellular structure from the confocal images: “... In ABAQUS, the cellular structure was constructed by extruding the cell outline to form the anticlinal walls, and the flat top and bottom periclinal walls were connected to the anticlinal walls using tie constraints. No material was assigned to the cell lumens to reflect the absence of turgor pressure.”

Section 4 Methods - Finite element simulations: “...The thickness of the epidermal cell wall of Arabidopsis leaves ranges from approximately 0.6 μm to 1.5 μm [94], so we set the thickness of the cell walls to 1 μm in our cellular structure model...”

“...We choose material parameters such that the initial modulus is equal to 1 and Poisson’s ratio is 0.3. The initial modulus can be arbitrarily scaled as it only affects the magnitude of forces. The initial Poisson’s ratio is chosen to fall within the experimental range...”

Section 3: “...In addition, we note that subcellular details are simplified in our FEM model, where we assume uniform thickness and homogeneous mechanical properties for both periclinal and anticlinal walls. Although we believe these simplifications do not substantially impact our main conclusions, they should be reconsidered if precise deformation patterns at the subcellular scale are of particular interest...”

For the five-beam model, we conceptualized the cellulose microfibril network as a simplified diamond-shaped five-beam structure with connectors, rather than using a detailed representation of the actual microfibril arrangement. The parameters for the five-beam model are: stretching stiffness k_s , bending stiffness k_b , initial length of the bending beam l_{b0} , initial tilt angle of the stretching beam is θ_0 , connector shear stiffness k_c , and connector sliding resistance D . The thickness of microfibrils is around 3.6 ± 1.9 nm (<https://doi.org/10.1073/pnas.95.12.7215>) and the adjacent distance between cellulose microfibril is around 20 – 150 nm (<https://doi.org/10.1111/j.1438-8677.1953.tb00272.x>). Based on these geometry estimates and beam theory, we estimated the stretching stiffness and bending stiffness ratio as 100. Accordingly, we set bending stiffness to 1, and stretch stiffness to 100. The initial length of the bending beam simply scales the force magnitude but does not qualitatively affect the

mechanical behavior, so we made an arbitrary choice for this length. We chose the initial tilt angle so that the initial incremental Poisson's ratio of the model was within the same range as the experiments. We chose connector shear stiffness and connector sliding resistance D so that the stiffening ratio and recovery ratio of the model were comparable to the experiments. The detailed explanation of parameters choices has been included in the **SI - Section B Five-beam model**. While this five-beam model captures key deformation modes, the actual structure of the cell wall is far more complex since it is irregular and multilayered. More realistic structural modeling may be required for precise quantitative predictions. These limitations have been discussed in **Section 3**.

The revised text is included below for reference:

SI - Section B Five-beam model: *"...The plant cell walls exhibit a characteristic diamond-shaped arrangement of cellulose microfibrils and the distance between adjacent cellulose microfibrils is between 20 to 150 nm [5]. The thickness of microfibrils is around 3.6 ± 1.9 nm [2]. So we chose the stretching stiffness and bending stiffness ratio $k_s/k_b = Al^2/3I \sim l^2/d^2 \sim 100$, where A and I represent the area and the second moment of the area. For a qualitative comparison, we chose the following parameters. We set $k_b = 1$ so that $k_s = 100$. The length of the beam simply scales the overall force-displacement response, so we made the arbitrary choice of $l_{b0} = 10$. The initial angle θ_0 will influence the incremental Poisson's ratio. We chose $\theta_0 = 1.3\pi/4$ so that the initial incremental Poisson's ratio of the model was within the same range as the experiments. We chose $k_c = 36$ and $D = 24$ so that the stiffening ratio and recovery ratio were comparable to the experiments on the 25d samples. The connector resistance was calculated as $C = 1/(1/D + 1/k_c)$. Different connector resistance values C_1 , C_2 , and C_3 were corresponding to three sets of chosen value (1) $k_c = 15$ and $D = 10$; (2) $k_c = 30$ and $D = 20$; (3) $k_c = 36$ and $D = 24$ respectively. Alternative initial angle values of $\theta_1 = 1.2\pi/4$ and $\theta_0 = 1.4\pi/4$ were chosen to demonstrate possible anisotropic effects."*

Section 3: *"...We conceptualized the arrangement of cellulose microfibrils as a simple diamond-shaped structure in the five-beam model. This model captures the key deformation modes of cellulose microfibrils: bending, reorientation, stretching, and slipping. It demonstrates that the observed three-regime nonlinear mechanical behavior arises from a transition between bending- and reorientation-dominated modes to stretching- and slipping-dominated modes. However, we note that the actual structure of the cell wall is far more complex since it is irregular and multilayered. If we imagine the wall as a distribution of diamond-shaped units with different sizes, each unit may undergo its bending-to-stretching transition at a different strain level. These local transitions may not only occur asynchronously but could also influence each other through network connectivity. In future work, we plan to incorporate more realistic cell wall architectures into our model to quantitatively predict mechanical responses..."*

The use of the *spr2-2* mutant is interesting, as its altered microtubule dynamics influence cellulose microfibril organisation and thus cell wall anisotropy. The anisotropy analysis supports the idea that changes in microfibril organisation affect stiffness. However, since *spr2-2* affects cell wall mechanics indirectly, it would be useful to include mutants that more directly impact cell wall synthesis, such as cellulose synthase mutants, or those that alter pectin modification. This would provide stronger evidence for the proposed link between microfibril connectivity and stiffness changes.

We thank the reviewer for the suggestion. Accordingly, we have conducted experiments using *qua2-1*, a mutant with around 50% reduction in homogalacturonan (HG) content compared with the wild type (Mouille et al, 2007, doi: 10.1111/j.1365-313X.2007.03086.x). We found that the initial stiffness (stretch from 1 to 1.05) and the early strain-stiffening behavior (stretch from 1.05 to 1.1) of epidermal peels were comparable to the wild type, while the stiffening regime was terminated earlier by softening and rupture (**Figure S7**). We think this result implies that HG reduction does not significantly impair early stiffening but compromises mechanical strength at higher strain, likely due to defective cell adhesion or loss of wall integrity. We have included these results and their interpretation in **Section 3**.

The revised text in **Section 3** is included below for reference:

*“... To assess whether reduced HG content affects mechanical behavior, we performed tensile tests on *qua2-1* (SI Methods E, Other plant material), a mutant with around 50% less HG and defective cell adhesion [90, 91]. We found that the initial stiffness (stretch from 1 to 1.05) and the early strain-stiffening behavior (stretch from 1.05 to 1.1) were comparable to the wild type. However, the stiffening regime was terminated earlier by softening and rupture (Figure S7). This suggests HG reduction does not significantly affect cellulose microfibril connections during early deformation but may compromise the strength of these connections and cell adhesion...”*

Figure S7

Figure S7: Comparison of monotonic tensile test results for wild type and *qua2-1*. Force per width – axial stretch and transverse – axial stretch curve for wild type at 25 days post-germination and *qua2-1* at 26 days post-germination. Note that five curves from wild type are reused from Figure 3B 25d data.

We also tested *rsw1-1*, a mutant of CESA1 involving cellulose synthesis, which exhibits reduction in cellulose production at high temperature (31 °C). We grew *rsw1-1* and wild type in a high temperature chamber (29.4 °C) (Figure R1). We found that mechanical behavior of epidermal peels is quite similar to wild type, or may be slightly less stiff than wild type (Figure R2). Previous research suggests that cellulose production of *rsw1-1* is reduced to half of wild type at 31 °C (<https://doi.org/10.1007/BF01280308>), which would be expected to result in a much softer mechanical response. We speculate that either the reduction in cellulose is not as pronounced at 29.4°C as at 31 °C, or cellulose density remains relatively unchanged. As a detailed investigation of this phenomenon is beyond the current scope, we plan to explore it further in future work.

Figure R1: (A) Wild-type plant and (B) *rsw1-1* mutant plant at 11 days post-germination, both grown in a 29.4 °C growth chamber.

Figure R2: Force per width (defined as membrane force in original submission) – axial stretch curves of leaf epidermal peel from the wild type and *rsw1-1* at 11 days post-germination.

Tracking mechanical properties in the context of plant development is a very interesting approach that is not that common in this type of papers, adding an important insight. However, a clearer description of the specific changes occurring in plant cell walls over time would strengthen the study. From the biological point of view, a more detailed account of how cell wall composition, microfibril connectivity, and organisation evolve and how these changes influence mechanical properties would make the developmental analysis more informative.

We thank the reviewer for the suggestions. We have expanded our discussion to include biological factors that may influence changes in mechanical properties during development in **Section 3**. Specifically, we have incorporated discussion on possible cell wall alterations during development, such as matrix polysaccharide integration into cell wall and pectin de-esterification.

The revised text is included below for reference:

“... In addition, there is evidence showing that matrix polysaccharide in the cell wall changes during development. As cells grow, newly secreted matrix polysaccharides, such as pectin and

hemicellulose, can diffuse into the wall and are integrated into the existing cellulose network. This integration is mediated by wall-modification enzymes and may enhance connectivity between cellulose microfibrils [91]. As cells mature, homogalacturonan(HG), a major pectin component, becomes less methyl-esterified. This enables more calcium-mediated cross-linking and results in stiffer pectin gels [92]....

Overall, the manuscript provides valuable insights into plant cell wall mechanics, but major revisions are needed. My recommendations to the authors are:

1. Strengthen the quantitative link between the theoretical model and the experimental data.
2. Expand the mutant analysis to include systems that more directly affect cell wall synthesis or pectin modification.
3. Provide a clearer discussion of specific developmental changes in cell wall structure and composition.

Finally, I would like to congratulate the authors. I believe this is a strong contribution to the field, and with some revisions, it has the potential to make a great impact.

We thank the reviewer for all the helpful suggestions. Each of the points has been addressed in our detailed responses to the individual comments above.

Reviewer #4 (Remarks to the Author):

In the present manuscript, the authors investigate the mechanical behavior of the leaf epidermis of the first true leaves of Arabidopsis through tensile testing. They use a series of complementary approaches to derive information on the behavior of the fibrous network material composing the cell wall. They focus in particular on the non-linearity of the stretching behavior and on Poisson's ratio, a property that describes the behavior of the material in a direction transverse to the load application. Through in silico modeling, the experimental results are interpreted and conclusions are drawn as to the role during plant developmental processes.

Overall, this is a very carefully executed study that is meticulously documented and provides important insight into the mechanical behavior of plant cell wall material in a growing organ. The insight is novel and the conclusions provide important new information.

I have a few concerns, however, which the authors may wish to consider.

We thank the reviewer for the thorough and thoughtful review. We appreciate the recognition of the novelty of our work. We have addressed the comments in the following response and accordingly in the revised version.

FUNDAMENTAL

1. Line 165: Determining Poisson's ratio of a material presumably requires for the flat strip of tissue to remain flat during axial load application. However, thin membranous material like the one tested here has the tendency to form wrinkles parallel to the load application to relax the compressive force resulting from the Poisson effect. Wrinkle formation presumably complicates the quantification of Poisson's ratio of the material property (vs the overall structure). Poisson's ratio of the epidermis material should be independent of the experimental setup of the tensile test, whereas the measured value for Poisson's ratio of a material wrinkling under tensile strain is presumably influenced by the exact setup and sample dimensions. It would therefore be crucial to validate whether wrinkles arise upon stretching of the samples examined here and in the specific setup and sample geometry used. This could be done by performing 3D confocal imaging on the stretched material and doing 3D reconstruction. Since wrinkle formation would potentially change the entire interpretation of a key message in this paper, I consider attention to this aspect a fundamental point that would need to be addressed.

We thank the reviewer for raising this important point. To address it, we performed 3D confocal imaging using a 20× air objective on the sample before and after tension. The 3D images confirmed that no significant wrinkling occurred during stretching. We have included this image as a new **Figure S2** and added a discussion in **Section 2.1** as follows: "...*We note that no*

obvious wrinkling was observed during testing (**Figure S2**), indicating that the deformation occurred primarily in-plane...”

Figure S2: 3D confocal images of a sample coated with fluorescent beads in undeformed (green) and deformed (yellow) states, showing no obvious wrinkling. (A) Isometric view; (B) side view along the sample width.

Related, Figure 1H suggests that the behavior of the material is visco-elastic. There seems to be little attention to this observation in the interpretation of the data. It could have significance for Poisson's ratio, since the ratio may show time-dependent behavior as well - and not necessarily in the same manner as the recovery from strain in axial direction. This highlights the importance of standard procedures (in terms of temporal behavior) when measuring Poisson's ratio in visco-elastic material. The authors may wish to consider this.

We thank the reviewer for this important point. We agree that the hysteresis observed in the curve indicates time dependent behavior of the material, and we have edited **Section 2.1** to address this. Comparatively, the Poisson's ratio shows relatively little time dependence in our tests. The corresponding transverse versus axial stretch curve of cyclic test shows that when reloading beyond previous maxima, the response closely aligns with the monotonic loading curve. Additionally, the recovery of transverse stretch is similar to that of axial stretch. We have included these plots in newly added **Figure S3** and added further discussion in **Section 2.1**.

For reference, the revised text is: “*The reloading curve is above the unloading curve, indicating the sample exhibits some viscous behavior in addition to plasticity. When the applied force exceeds the maximum force previously applied in the incremental cyclic test, the force per width versus the axial stretch response (Figure 1H) follows the response from the monotonic tensile tests (Figure 1D). To quantify how much deformation is recoverable during stretching (the ability to return back to the original size), we calculated the recovery percentage (Figure 1I). The*

recovery versus axial stretch curve also exhibits a three-regime behavior. Most of the deformation (more than 80%) is recoverable in regime I, the recoverable deformation sharply decreases from 80% to 40% in regime II, and then eventually levels off in regime III. The transverse–axial stretch response under cyclic loading also largely follows the monotonic behavior (Figure S3A). Unloading happens with reduced transverse strain recovery, and unlike for the axial stress-strain response, there is relatively little hysteresis between unloading and reloading. The recovery of transverse stretch shows a similar magnitude and trend to that of axial stress (Figure S3B).”

Figure S3: Transverse stretch change from the same incremental cyclic tensile test shown in Figure 1H. (A) Transverse stretch versus axial stretch curve. (B) Recovery of transverse stretch and axial stretch, each plotted against axial stretch to compare their respective behaviors.

2. Interesting results regarding the anisotropy of the tensile behavior of the epidermis were made in the mutant *spr2-2*. In the discussion, the mechanical anisotropy is related to the 'enhanced alignment of microfibrils' (line 562) which is reasonable but no proof is provided. I suggest that experimental evidence for the same be added to the manuscript. Standard staining protocols for cellulose are readily available (e.g. Bidhendi et al 2020. *Journal of Microscopy* 278: 164-181) and would significantly strengthen the Conclusions.

We thank the reviewer for this thoughtful suggestion. We agree that direct evidence of cellulose orientation would strengthen our conclusions. However, due to these technical limitations, we were unable to obtain reliable orientation data in a timely manner. We attempted to visualize cellulose microfibril orientation using calcofluor white staining and confocal microscopy, following the protocols of Bidhendi et al. (2020). However, we encountered reproducible imaging artifacts likely caused by polarization-dependent fluorescence. Specifically, cellulose fibrils consistently appeared aligned in the horizontal direction in confocal images, regardless of which

samples were imaged or how the samples were physically oriented, as shown in the following figures. This artifact was observed under both 40× water and 63× oil immersion objectives. We believe this directional bias is due to the cell wall bifluorescence (fluorescence dependent on the polarization of the excitation light, as observed by Thomas et al. 2017, doi: 10.1111/jmi.12582) for calcofluor white stain. Since the confocal lasers are linearly polarized in the horizontal direction, fibrils aligned with the laser polarization fluoresce more strongly, while those in other orientations are poorly visualized. Thomas et al. 2017 (doi: 10.1111/jmi.12582) demonstrates that excitation polarization influences apparent fibril orientation on confocal images. While Thomas's work has suggested rotating either the sample or the scanning field to analyze orientation of microfibrils, we found this approach impractical in our case due to rapid photobleaching of the dye after repeated scans. Alternative cellulose dyes Pontamine fast scarlet 4B and Congo red also have this bifluorescence.

Figure R3: Red line within each cell represents the net orientation of its cellulose microfibrils obtained by using fibrilTools (10.1038/nprot.2014.024) analysis in MorphographX(<https://morphographx.org/>). In the original orientation (left), most red lines align with the horizontal direction. After rotating the same sample by 90° (right), the red lines still predominantly align with the horizontal direction, indicating an imaging artifact.

Figure R4: Magnified images from Figure R3. Microfibrils are visible as thin white striations within each cell. Apparent cellulose microfibrils in the original orientation (left) are predominantly aligned in the horizontal direction. After rotating the same sample by 90° (right), the microfibrils still appear mostly horizontal, further indicating this imaging artifact.

MAJOR

3. While much of the relevant literature is cited, the authors may wish to consider referring to Rosemary Dyson's work, e.g. doi.org/10.1016/j.jtbi.2021.110736 (in the context of page 8) or this book chapter (<https://escholarship.mcgill.ca/concern/articles/q811kr12s>)

We thank the reviewer for the recommendation. We have read the suggested paper, book chapter and other work by Rosemary Dyson, and have incorporated them into the revised manuscript. We have cited the book chapter in the first paragraph of **Section 1**. We have cited these two papers "*Lockhart with a twist: modelling cellulose microfibril deposition and reorientation reveals twisting plant cell growth mechanisms*" and "*A fibre-reinforced fluid model of anisotropic plant cell growth*" in the fourth paragraph of **Section 1**.

4. Line 140: From this description it appears that the epidermal peel consists of the intact epidermal cell layer meaning that it comprises the upper and lower periclinal walls and the anticlinal walls, and that the epidermal cells are intact (albeit deflated). If this is correct, this fact should be spelled out very clearly since in the much used onion epidermal peels used by others, the peel consists only of half of the epidermal layer since during stripping, the anticlinal walls

rupture (whereas in the present manuscript the rupture seems to be at the connection between mesophyll and epidermis).

In the same context: Was it consistently the abaxial epidermis that separated when performing the manoeuvre shown in Fig 5? This might be worth mentioning.

We thank the reviewer for this helpful comment. Yes, the epidermal peel comprises an intact epidermal cell layer, including both upper and lower periclinal walls as well as the anticlinal walls. We have clarified this point in **Section 2.1**. In addition, we conducted Z-stack imaging of an epidermal peel and included it as a new **Figure S1B** to show its 3D structure more clearly. Yes, the abaxial epidermis is consistently separated, as shown in Fig S5. We have now clarified this in the **Method - Leaf epidermal peel and tensile test sample preparation**.

The revised text is included below for reference:

Section 2.1: “...Each peel has one layer of abaxial (bottom side of the leaf) intact epidermal cells, which comprises both the upper and lower periclinal walls as well as anticlinal walls (Figure S1)...”

Method - Leaf epidermal peel and tensile test sample preparation: “...We note that it’s consistent that abaxial epidermis is removed from the leaf...”

B

Figure 1B: 3D confocal images of a sample stained by calcofluor white in isometric view and side view.

5. Also, here and elsewhere it would help if the term 'periclinal wall' was used throughout where appropriate.

We appreciate this suggestion and have revised the manuscript to use the term periclinal wall wherever appropriate.

6. Line 146: I find the term 'membrane force' an unfortunate choice in the present context. I understand that it is probably taken from the engineering language, but for a biologist, the term 'membrane' has such a dominant, very different significance that it keeps distracting me. While it is up to the authors, I suggest finding a different term (sheet force?).

We have discussed this suggestion and to improve clarity and avoid confusion, we have replaced "membrane force" with "force per width" throughout the manuscript.

7. Line 146: What the authors define to be the 'axial stretch' is really an axial 'strain' in engineering terms. I suggest using that terminology instead.

We have clarified our terminology in **Section 2.1**: in our study, "stretch" refers to the ratio of deformed to initial length, which is commonly used in large-deformation mechanics.

The revised text is included below for reference: "...axial stretch (deformed length of the sample divided by the initial length of the sample, a convention for large-deformation mechanics)..."

8. Line 150: Is the test strip truly under tension in Regime I or is it relaxing tissue folds (oriented transverse to the load application) that might have been created during the excision process?

We thank the reviewer for the comment. We confirm that the test strip is truly under tension in regime I. We provided 3D confocal images showing the initial flat status of the strip in a newly added **Figure S2**.

9. Line 245: What are the dimensions or properties of the 'single plate'? A plate with thickness (or stiffness) twice that of a single periclinal wall plate in the cellular model? Please specify.

We thank the reviewer for this question. The "single plate" has the same mechanical properties as a single periclinal wall. Its thickness is set to twice that of the periclinal wall to match the total thickness of periclinal walls in the cellular model. We have clarified this in **Methods - Finite Element Simulation** and referred to it in the main text.

The revised text is included below for reference:

Method - Finite Element Simulation: *“For the single plate model, the thickness was set equal to the combined thickness of the two periclinal walls in the cellular model, i.e., 2 μm , 1.2 μm , and 4 μm respectively.”*

10. Line 249 and Fig 2B: Neither from the figure nor the text is it truly clear how the FE model is structured. Do the anticlinal walls connect to the outer and inner periclinal sheets? Is the lumen of the cells empty? Does the lumen have properties?

Also, was the fact that in the biological system the anticlinal walls are thinner than the periclinal walls of the epidermis, and that the out periclinal wall is thicker than the inner periclinal wall, considered in the model? I don't think it would make much of a difference in the outcome here, but it would truly be helpful to specify the simplifying assumptions much more clearly, certainly where they deviate from biological reality.

Thanks for pointing out this gap in provided information. We have added a more detailed explanation of the FEM model in **SI Methods - Extracting the cellular structure from the confocal images**, including how anticlinal walls are connected to periclinal walls via tie constraints and how the lumen is modeled as empty to reflect the absence of turgor pressure. The wall thicknesses are assumed to be uniform (now stated in **Methods - Finite Element Simulation**). For our revised manuscript, we have also verified the loss of turgidity by conducting frozen and thawed treatment experiments, which has been included in **Section 2.2**. We have also added discussion of modeling assumptions in **Section 3**.

The revised text is included below for reference:

SI Methods - Extracting the cellular structure from the confocal images: *“... In ABAQUS, the cellular structure was constructed by extruding the cell outline to form the anticlinal walls, and the flat top and bottom periclinal walls were connected to the anticlinal walls using tie constraints. No material was assigned to the cell lumens to reflect the absence of turgor pressure.”*

Section 2.2, *“...After peeling, the epidermal cells lose their turgor pressure (Figure S1). We further verified this by performing tensile tests on the frozen and thawed treatment sample (SI Method - Leaf epidermal peel preparation), and we showed mechanical behavior similar to that of untreated samples (Figure S1C). Therefore, in the cellular structure model, the top and bottom periclinal cell walls were treated as flat plates...”*

Methods - Finite Element Simulation: *“...For simplicity, we assumed the same thickness for all periclinal and anticlinal walls in the model.”*

Section 3: “...In addition, we note that subcellular details are simplified in our FEM model, where we assume uniform thickness and homogeneous mechanical properties for both periclinal and anticlinal walls. Although we believe these simplifications do not substantially impact our main conclusions, they should be reconsidered if precise deformation patterns at the subcellular scale are of particular interest...”

11. Line 262: I suggest adding ‘...from the material PROPERTIES of the PERICLINAL cell walls’. If this is not what is meant here, please explain the statement better in different manner.

Thanks for the suggestion, and we have modified the sentence as suggested.

12. Line 274: While the microfibrils in the periclinal walls of the Arabidopsis leaf epidermis are indeed isotropically oriented IN AVERAGE or OVERALL, in these wavy pavement cells there is significant complexity at subcellular level with instances of anisotropic orientation (Altartouri et al 2019 Plant Physiology 181: 127–141). This fact should at least be mentioned.

We thank the reviewer for this important point. We have explicitly acknowledged the presence of anisotropic orientation at subcellular level in pavement cells in the revised version in **Section 2.2**.

The revised text is included below for reference:

Section 2.2: “This observation implies that the cellulose microfibrils are initially distributed evenly in all orientations on average at the tissue level. We note that the guard cells, in which the cellulose microfibrils are highly aligned, appear brighter (Figure 2D). In addition, there is brightness fluctuation within each cell, which is due to subcellular regions of anisotropic cellulose alignment in wavy pavement cells [54].”

13. Line 276: This is a very well known effect, and in the context of plant cell walls has been proposed as early as 1953 (multinet growth theory by Roelofsen and Houwink (1953) Acta Botanica Néerlandica 2:218–125). Mention of this would be appropriate in order to give credit to early thinkers.

We thank the reviewer for the suggestion. We have now explicitly mentioned the multinet growth theory and discussed the consistency between our result and the theory in **Section 2.2**.

The revised text is included below for reference:

Section 2.2: “...*This behavior resembles the reorientation of cellulose microfibrils toward the axial direction during unidirectional cell growth, as described by multinet growth theory [55]...*”

14. Lines 416-417: Although it is not intended this way, this sentence may suggest to non-expert readers that plant cell growth is driven by an increase of turgor. That notion would be incorrect, of course. The trigger for growth is a reduction in cell wall stiffness instead, leading to a drop in turgor that in turn leads to an uptake of water that stretches the wall. Maybe the statement could be reworded somewhat.

We thank the reviewer for the valuable comments. We agree the original sentence may cause misunderstanding, and we have reworded it as follows:

“This result implies that older cells tend to accumulate less plastic deformation than younger cells at the same level of turgor pressure.”

MINOR

15. Lines 81 & 315: 'It's' is inappropriately informal and should be 'it is'

We have corrected “it’s” to “it is” for both instances.

16. Line 139: Although it is mentioned in the Methods section, I suggest mentioning here that exclusively the first true leaves are used for testing. Also, the orientation and location of the standard test strip should be described or, even better, shown as a drawing or photograph.

We now explicitly mention that the first true leaves are used for testing in section 2.1: “*we stretched the epidermal peels of first pair of true leaves from Arabidopsis*”

We have also added a description of the orientation and location of the test strip to the same section: “*We cut a thin rectangular strip for testing, with the long side parallel to the midrib and positioned approximately midway between the midrib and the leaf margin.*” And we illustrated it in Figure 1A.

17. Figure legend 1C: Specify whether these are single optical sections or projections of a 3D stack.

We thank the reviewer for this suggestion. The undeformed image is a projection of a 3D stack, while the deformed image is a single optical section. We have clarified this in the legend for Figure 1C.

18. Lines 253 and 256 I suggest adding '*...deviation WITHIN THE TESTED RANGE OF DEFORMATION*'

We thank the reviewer for the suggestion. We have added “within the tested range of deformation” for both sentences.

Reviewer #1 (Remarks to the Author):

We thank the reviewer for reviewing our revised manuscript and our responses and for helpful comments.

Still one point must be improved:

The explanation regarding my comment on “using N/m instead of the stress units (Pa)” is speculative and hard to understand. Following the explanation, I agree that units in the equations are ok. But, the explanation refers to relation of turgor (in single cell) to force membrane (of the tissue) while cells are not turgid in this case (specific experiment to confirm this fact has been added). To clarify your way of thinking, I suggest to present detailed explanation related to this problem in Supplementary Materials. On the other hand, it is very intuitive that in the experimental stretching the sample cross-section (not only width) determines the force detected on the sensor. I must admit that I do not have experience in studying epidermis and my suspicious about varying thickness related to development stage and mutant is maybe not very important. But, if the thickness of the stretched stripe changes substantially, it would not mean that the material property (stiffness) of the epidermis changes at all while the measured force-strain change results from change of thickness. Therefore, I consider my question and authors explanation regarding this point as fundamental for this work (hopefully explained in details in supplementary and more clearly than in the response to my request).

We thank the reviewer for the thoughtful follow-up and for suggesting a detailed explanation in the Supplementary Materials. To address the reviewer’s concern, we have included a detailed explanation in the added Supplementary Methods section “Choice of force per width as a measurement unit”. Deformation as a function of force per unit width can be convenient to apply without acquiring cell wall thickness. One important example is modeling the relation between turgor pressure and cell wall deformation in a turgid cell. The relation is: $\text{turgor pressure} = \text{stiffness} * (\text{stretch} - 1) * \text{perimeter of cross section} / \text{cross-sectional area of the cell}$. We note that the use of stiffness in N/m is only valid when there is no substantial variation in wall thickness across samples. Plant primary cell wall thickness stays relatively constant as the cell grows in most cases, since the wall synthesis and wall expansion are well-coordinated (see e.g. Lockhart 1965; Cosgrove 2022). However, if significant thickness variation across samples is present, the use of stiffness in N/m may lead to misinterpretation. We have included these points in the added section.

Choice of force per width as a measurement unit

In this study, we report force per unit width (N/m) rather than stress (Pa) for the following reasons. Plant primary cell wall thickness stays relatively constant in most

cases as the cell grows, since wall synthesis and wall expansion are well-coordinated (Lockhart 1965; Cosgrove 2022). This suggests that samples in our tests have similar wall thickness, so force per width reflects intrinsic mechanical stiffness. In addition, the use of force per width avoids the need to measure wall thickness directly, which may be impractical to measure consistently since the cell wall is extremely thin and its thickness varies within a single cell. Furthermore, force per width (N/m) allows a direct connection between experimental measurements and the biophysical model that relates turgor pressure to cell wall deformation (Jensen 2023) without requiring explicit thickness measurements. From tensile tests, we obtained how much the cell wall deforms under a given level of force per width, captured by the force per width-stretch relationship: force per width = stiffness * (stretch -1). This relation can be used to relate turgor pressure to cell wall deformation in a turgid cell. Within a cell, the balance of forces gives turgor pressure = stiffness * (stretch -1) * perimeter of cross section/ cross-sectional area of the cell, without acquiring explicit thickness. In general, we note the use of stiffness in N/m is appropriate to understand the intrinsic cell wall material properties only when there is no substantial variation in wall thickness across samples.

Reviewer #2 (Remarks to the Author):

All comments have been adjudicated. I sincerely thank the authors for their thorough and thoughtful responses.

We thank the reviewer for the comments and helpful feedback during the reviewing process.

Reviewer #3 (Remarks to the Author):

This revised manuscript addresses the key points raised in my previous review and improves the clarity and biological relevance of the findings. Overall, the authors have responded thoroughly. The revised manuscript presents a more complete account of how wall composition and architecture relate to nonlinear mechanics in epidermal tissues. The modelling, mutant analysis, and developmental comparisons are now more clearly connected.

In response to my and other reviewers' comments, the authors have strengthened the connection between their theoretical models and experimental data. The finite element simulations are now more clearly described, including how cell geometries were extracted and how turgor loss and wall thickness were handled. The assumptions about wall structure and material properties are now explicitly stated. While the model still simplifies subcellular features, the authors have acknowledged its limitations and clarified its intended use.

The five-beam model is also better justified. Parameter choices are supported by literature-based estimates of cellulose microfibril geometry and spacing, and the model's capacity to capture distinct deformation modes is now more clearly explained. The authors have added a discussion of the model's simplifying assumptions, as requested.

My earlier suggestion to include additional mutants affecting wall synthesis or matrix composition has been addressed. The addition of *qua2-1*, which reduces homogalacturonan content, is appropriate and informative. The mechanical results from *qua2-1* show that early stiffness is unaffected, while rupture occurs earlier than in wild type, consistent with compromised cell adhesion or wall cohesion. This strengthens the conclusion that pectin contributes more to strength at high strain than to initial stiffness. Although the *rsw1-1* data are not included in the manuscript, I appreciate the authors sharing the results in the rebuttal. The mechanical similarity to wild type at the tested temperature is interesting and may warrant further study, though it is reasonable to leave this outside the scope of the current manuscript.

We thank the reviewer for reviewing our revised manuscript and our responses. We thank the reviewer for the positive feedback regarding our revision.

The discussion of developmental changes in wall composition has been expanded to include relevant biological processes, such as pectin de-esterification and matrix polysaccharide integration. These additions make the developmental aspect of the study more informative. However, the term "microfibril connectivity," used to explain the increase in stiffness over time, remains broad. It would be helpful if the authors could clarify whether this refers to tighter bundling, greater inter-fibril adhesion,

matrix cross-linking, or another structural feature. Even a conceptual definition would help ground the interpretation.

We thank the reviewer for this comment. We have replaced “connectivity” as “physical connections”, and clarified what mechanisms may contribute to these changes in the **Discussion** section. The revised sentence is provided below.

“During growth, newly secreted matrix polysaccharides, such as pectin and hemicellulose, are integrated into the existing cellulose network by enzymatic and spontaneous crosslinking mechanisms⁹², and such crosslinking may enhance physical connections between cellulose microfibrils”

It might also be useful to consider whether the observed changes in plasticity or recovery could provide indirect support for reduced slippage, if no direct structural evidence is available. Additionally, it would be helpful to note whether the findings generalise beyond the sampled leaf ages and positions, or if they are limited to the specific developmental stages examined.

Yes, we do think observed change in plasticity or recovery could provide indirect support for underlying structure changes. In our experiment, we showed that recovery of samples is reduced and samples become more plastic as samples deform, indicating that slippage between fibers is increasing. Eventually, recovery reaches a plateau, suggesting fiber stretching mode dominates and the deformation contributions from fiber-fiber slippage and elasticity have stabilized. This point is further supported by the five-beam model. We have now added an explanation of the last plateau stage in the last paragraph of **Section “Nonlinear mechanical behavior comes from the cell wall - a fibrous network material”**.

Regarding generalizability, our main finding, that the cell wall exhibits fibrous network-like mechanical behavior with high tunability, is likely to extend beyond the leaf ages and position from within the plant, as the cell wall composition and so the underlying structure mechanisms are expected to be similar. However, further work is needed to confirm their applicability across broader developmental stages, different leaf positions within a plant, and across different plant organs. We have added this point in our **Discussion Section**.

The revised text is included below for reference

Section “Nonlinear mechanical behavior comes from the cell wall - a fibrous network material”: “...Eventually, recovery reaches a plateau, suggesting fiber stretching mode dominates and the deformation contributions from fiber-fiber slippage and elasticity have stabilized...”

Discussion Section:“...We believe that viewing the cell wall as a fibrous material, where mechanical behavior arises from the collective motion of cellulose microfibrils and exhibits a high degree of tunability, can be extended beyond the specific leaf ages and positions within plants examined in this study. This perspective may apply more broadly to primary cell walls, as their main structures are similar to each other and consist of cellulose microfibrils embedded in a pectin and hemicellulose matrix. However, further work is needed to confirm its applicability across a broader range of developmental stages, leaf positions within plants, organs, and plant species.”

In addition to addressing my own comments, the authors have incorporated relevant revisions in response to points raised by the other reviewers. These include clearer treatment of anisotropy and imaging limitations, justification of force units, clarification of modelling terminology, and a more explicit discussion of assumptions. These revisions further strengthen the manuscript and improve its clarity.

We thank the reviewer for reviewing revisions in response to other reviewers and we appreciate helpful feedback from the reviewer.

Reviewer #4 (Remarks to the Author):

The authors have addressed or rebutted most of my previous comments satisfactorily. It is a fantastic work, congratulations.

We sincerely thank the reviewer for many helpful comments and suggestions, and we truly appreciate the positive recognition from the reviewer.

A few minor concerns remain:

Lines 270, 424 and potentially elsewhere: I think that typically, there would be no article ('the') before Poisson's ratio, similar to, say, 'Hooke's law'.

We have deleted "the" in all six places we found it.

Line 756: 'It's' is inappropriately informal and should be 'it is'.

We have corrected it.

The authors omitted addressing or rebutting two of my previous comments:

We apologize that we overlooked these two comments by mistake. We have addressed these comments this time.

19. Line 458 (now line 532): I suggest specifying the plant organ that was investigated in the study by Hervieux et al.

We have specified that the sepal was investigated in Hervieux's paper.

20. Line 574 (now line 645): I suggest adding '...different organs, species AND GROWTH STAGES'

We thank the reviewer for the suggestion and we have added "and growth stages" in that sentence.